# Mitigation of Breast Cancer Cells’ Invasiveness via Down Regulation of ETV7, Hippo, and PI3K/mTOR Pathways by Vitamin D3 Gold-Nanoparticles

**DOI:** 10.3390/ijms25105348

**Published:** 2024-05-14

**Authors:** Moumita Roy, Fazle Hussain

**Affiliations:** Mechanical Engineering Department, Texas Tech University, Lubbock, TX 79409, USA

**Keywords:** VD3-GNP treatment, PI3K, AKT, mTOR, Hippo, ETV, pathway

## Abstract

Metastasis in breast cancer is the major cause of death in females (about 30%). Based on our earlier observation that Vitamin D3 downregulates mTOR, we hypothesized that Vitamin D3 conjugated to gold nanoparticles (VD3-GNPs) reduces breast cancer aggressiveness by downregulating the key cancer controller PI3K/AKT/mTOR. Western blots, migration/invasion assays, and other cell-based, biophysical, and bioinformatics studies are used to study breast cancer cell aggressiveness and nanoparticle characterization. Our VD3-GNP treatment of breast cancer cells (MCF-7 and MDA-MB-231) significantly reduces the aggressiveness (cancer cell migration and invasion rates > 45%) via the simultaneous downregulation of ETV7 and the Hippo pathway. Consistent with our hypothesis, we, indeed, found a downregulation of the PI3K/AKT/mTOR pathway. It is surprising that the extremely low dose of VD3 in the nano formulation (three orders of magnitude lower than in earlier studies) is quite effective in the alteration of cancer invasiveness and cell signaling pathways. Clearly, VD3-GNPs are a viable candidate for non-toxic, low-cost treatment for reducing breast cancer aggressiveness.

## 1. Introduction

### 1.1. Breast Cancer

Breast cancer is caused by numerous factors, e.g., age, hormonal imbalance, BRCA mutations, etc., and is one of the leading causes of death in women worldwide [1,2]. It is a heterogeneous disease classified under the following various subtypes: (i) luminal A and B (accounting for 50–60% of breast cancer cases); (ii) basal-like or triple-negative (TNBC) (10–20%), which lack the ER, PR, and HER2 receptors; and (iii) HER2 (human epidermal growth factor receptors, 10–15%) [3]. The extent of the expression of hormone receptors on the breast cancer cells’ surfaces acts as a marker of aggressiveness. As an example, estrogen-receptor positive cancer cells (ER+) (e.g., MCF-7) are less aggressive than TNBC and are easier to treat. On the other hand, TNBC (MDA-MB-231), prevalent among black women [4], lacks any cell surface markers and, hence, is difficult to treat. A recent clinical study using tarstuzumab and deruxtecan in low-HER2 positive breast cancer [5] is a promising primary treatment in cancer relapse.

Metastasis, the spread of cancer cells from a primary tumor to distant organs, is the leading cause of cancer deaths [6]. Traditional cancer therapies rarely work with TNBC, due to the lack of appropriate breast cancer cell surface receptors [5]. Conventional chemotherapy often leads to severe toxic side effects [7]. With the goal of developing a non-toxic therapy (see later), we focus on mitigating breast cancer invasion and the related cellular pathways and we achieve our goal using VD3-GNPs (where VD3-PEG is conjugated with the GNPs).

### 1.2. PI3K/AKT/mTOR

The PI3K/AKT/mTOR pathway (Figure 1) is a master regulator of cancer cell signaling, which leads to cell growth, invasion, migration, apoptosis, and glucose metabolism [8,9,10]. This pathway has the following three major components: (i) phosphatidylinositol 3-kinase (PI3K), (ii) a downstream molecule serine/threonine-protein kinase B (PKB; also known as AKT), and (iii) mTOR complexes (mTORC1, mTORC2—which constitute a feedback loop for AKT activation—and mTORC3) [8,9,10]. The phosphorylation and dephosphorylation of the regulatory proteins control this pathway, by modulating the levels of protein expression [11]. mTOR signaling is often overactive in cancers and the different isoforms act as key cancer regulators [12]—mTORC1 regulates protein synthesis and autophagy, mTORC2 regulates kinases of the AGC family, and mTORC3 causes chemotherapy resistance, though its specific role in breast cancer is yet to be understood [13]. Hence, the therapeutic regulation of the PI3K/AKT/mTOR pathway can suppress breast cancer [12,13].

### 1.3. ETV7

ETV7 is a member of the ETS (E26 transformation-specific) family of transcription factors. Its overexpression is associated with multiple cancer-causing mechanisms, including tumorigenic transformation; oncogenic progression; and genetic regulation for development, differentiation, proliferation, migration, and apoptosis [14,15,16,17,18]. ETV7 also plays a bimodal role in mTORC1/2 activity and is mTOR’s binding partner (in mTORC3) [19,20,21]. Here, we explore, for the first time, the role of VD3-GNP treatment on the PI3K/AKT/mTOR cascade, including the regulation of ETV7.

### 1.4. Hippo

Hippo signaling acts on several biological processes, e.g., organ development, cancer progression, epithelial–mesenchymal transition (EMT), and tumor development [22,23,24,25,26]. Two important transcriptional co-activator components of the Hippo pathway, YAP (Yes-associated protein or YAP1) and TAZ (transcriptional co-activator with PDZ-binding motif), shuttle between the nucleus and cytoplasm and regulate gene expression by binding with the TEAD (TEA/ATTS domain) family of transcription factors (TEAD1–4) within the nucleus and control their function [22,23,24,25,26]. The kinases mammalian sterile 20-like kinase (MST) and large tumor suppressor (LATS) are successively phosphorylated and regulate YAP/TAZ shuttling between the cytosol and nucleus [22,23,24,25,26] (Figure 1).

Extensive interaction between the Hippo pathway and PI3K/AKT/mTOR promotes cell proliferation, migration, and aggressiveness in breast cancers; induces miR-29 to inhibit PTEN; and promotes the PI3K-mTOR pathway to regulate tissue growth and hyperplasia [27]. In pulmonary hypertension, HIPPO/LATS1, self-supported via a Yap–fibronectin–ILK1 signaling loop, is a newly recognized mechanism of self-sustaining proliferation [28]. YAP/TAZ activates TEAD, to induce expression of the high-affinity leucine transporter LAT1, which is a heterodimeric complex of amino acid transporters (SLC7A5 and SLC3A2) [29,30]. These play an essential role in amino acid-induced mTORC1 activation.

### 1.5. Vitamin D3-Gold Nanoparticles

Breast cancers are highly complicated and a variety of therapeutic strategies have attempted to treat them, e.g., therapeutic alteration of pathway components, applications of immune checkpoints, androgen inhibitors [31], etc. Here, we study the role of Vitamin D3-gold nanoparticle treatment in the alteration of the Hippo pathway in ER (+) (MCF-7 cells) and TNBC (MDA-MB-231 cells).

The prohormone Vitamin D (VD) is a precursor to its biologically functional metabolite—a lipophilic seco-steroid hormone known as calcitriol (VD3)—that plays a critical role in calcium, magnesium, and phosphate metabolism [31,32,33]. Research has identified VD as a breast cancer risk factor; a circulating VD level ≥ 45 ng/mL presumably protects against breast cancer [31,32,33]. Other studies revealed an inverse correlation between VD level and breast cancer risk [31,32,33,34,35,36]. Stoll et al. [34] found that elevated serum VD3 (through sun exposure and dietary intake of more than 400 IU per day) significantly decreases breast cancer incidence and recurrence, while serum VD3 deficiency can lead to breast cancer occurrence. So, these findings indicate that using VD in cancer is highly promising, but its dosage and delivery are important, because high doses are toxic [34].

Unfortunately, excessive VD intake is not rare and can cause severe toxicity and hospitalization [32,33,34,35,36,37,38,39,40,41,42,43]. The challenges in VD deficiency and its therapy are its low bioavailability [36], high degradation in systemic circulation [37], and toxicity (VDT) at high dosage, which can lead to hypervitaminosis D symptoms and hypercalcemia [36,37].

Lipophilic VD3 is a treatment challenge, as an aqueous body environment cannot absorb it [44,45,46]. GNPs have multiple benefits over other nanomaterials, due to their unique physicochemical properties. The GNP’s surface can easily incorporate ligands, including functional groups such as thiols, phosphines, amines, and hormones like VD3. The advantage of VD3 delivery using gold nanoconjugates lies in its increased cellular uptake via endocytosis, which increase (~40%) its half-life in systemic circulation [47,48,49,50,51,52,53] by bypassing the body’s metabolic pathways. Regular VD3 mostly degrades during cellular metabolism and is often stored in adipose tissues, which causes its very low bioavailability. Our approach of using gold nanoparticle-mediated VD3 therapy can easily overcome the problems mentioned above—low systemic circulation, low lifetime, and toxicity for high dosages [47,48,49,50,51,54,55].

### 1.6. Objective

The objective of our study is to evaluate whether vitamin D3 conjugated to gold nanoparticles—VD3-GNPs—can reduce breast cancer aggressiveness and alter important cancer cellular pathway proteins (PI3K/AKT/mTOR, ETV7, and YAP/TAZ), so that the VD3-GNPs can be a cancer therapy.

## 2. Results

### 2.1. Characterization of VD3-GNPs

#### 2.1.1. TEM Images

TEM images of VD3-GNPs (Figure 2A,B) show spherical morphologies, while the GNPs (Figure 2C,D) have both spherical and elliptical shapes. A VD3 corona is visible outside the VD3-GNPs, when compared with GNPs (at magnification ~200,000× (Figure 2E–H). The quantification of TEM revealed that the average diameter of VD3-GNPs = 16.6 ± 2.41 nm and of GNPs = 26.11 ± 9.02 nm. The size distributions of VD3-GNPs and GNPs from TEM images are shown in Figure 2I,J.

#### 2.1.2. UV–Visible Spectrum

The UV–Vis spectra of the particles are shown in Figure 3A. The maximum absorbance in GNPs and VD3-GNPs was observed at 526 and 530 nm, respectively. The spectral shape was similar, but shifted to a 3–4 nm longer wavelength for VD3 conjugation with GNPs.

#### 2.1.3. Evaluation of Vitamin D3 Coupling to Gold Nanoparticle Surfaces Using FT-IR

To identify the molecular distribution of the vitamin D3 and VD3-GNPs in the molecular scaffold of the gold nanoparticles, we recorded the infrared absorbance spectra using a Fourier Transformation Spectrometer (Bruker Scientific, LLC, Billerica, MA, USA) (Figure 3B). In Figure 3B, for VD3-GNPs (blue), GNPs (green), and VD3 (red), the bands at 1750 cm^−1^ correspond to the asymmetric and symmetric stretching of C=O^−^. The 3650–3200 cm^−1^ range, with a peak close to 3300–3200 cm^−1^, reveals the -OH stretch of the phenol group of VD3-GNPs and VD3. The FT-IR spectrum peak at 1650–2000 cm^−1^ corresponds to the C-H bending of the aromatic carbon ring of the VD3-GNPs, as well as VD3. The free thiol group on the PEG-VD3 surface helps to conjugate the VD3, via thiolate bonds with GNPs. Therefore, the broad band in the 2400–2000 cm^−1^ region corresponds to the thiol (S-H) bond of the VD3-GNPs (Figure 3B). The FT-IR spectra of VD3 (cholecalciferol) is characterized by the presence of an O-H stretching peak at 3300–3200 cm^−1^, Sp^3^ C-H (CH3 and CH2) asymmetric stretching bands at 2943 and 2875 cm^−1^, a C¼C stretching band ~1680 cm^−1^, and a C-O-C stretching band at 1162 cm^−1^, respectively.

#### 2.1.4. DLS

The distribution of the nanoparticle mixture diameter, as obtained using DLS, showed that VD3-GNPs in solution have a mean diameter of ~34.6 ± 0.6 nm (see Appendix A, Table 1). Note that these hydrodynamic diameters are ~2.62-fold larger than those measured using TEM (~16 nm); this is expected as TEM is performed on dry samples and DLS is performed on samples diluted in an aqueous phase (due to hydrogen bonding with water molecules in the aqueous solution).

#### 2.1.5. Zeta Potential Measurement

The Malvin zeta track analyzer revealed the zeta potential to be P_avg_ = 24.4 ± 1.8 mV for VD3-GNPs and, for GNPs, P_avg_ = −17.5 mV ± 2.0.

#### 2.1.6. HPLC Quantitation of VD3 in VD3-GNPs

Vitamin D3 levels in the VD3-GNPs were quantitated using high-performance liquid chromatography (HPLC); we used 1, 2.5, 5, and 10 mM Vitamin D3 standards (the data for 10 mM are shown in Figure 4A). The HPLC elution of Vitamin D3 peaks at 2.8 min for both the standard and the VD3-GNPs (marked by the arrow). The HPLC analysis suggests that the VD3 concentration of the 100 µL VD3-GNP sample is 388 nM. For our subsequent cell-based studies, we use 10 µL from our 388 nM stock of VD3-GNPs, which yields 38 nM VD3 (see Figure 4B).

### 2.2. Cell-Based Studies

#### 2.2.1. MTT Assay after VD3-GNP Treatment

MDA-MB-231 and MCF-7 cells (cultured for 24 h) were treated with VD3-GNPs for another 24 h and the control groups were left untreated in a 5% CO_2_ incubator at 37 °C. The control and VD3-GNP concentrations (10 µg/mL, 1:10, 1:100, and 1: 1000), respectively, show 100%, 100.45%, 106%, 110%, 99%, 100%, 102%, and 104% cell survival (non-significant statistical difference); see Figure 5A,B. The 10 µg/mL concentration was used for further studies, which has 38 nM VD3, as obtained from our HPLC data.

#### 2.2.2. Cell Invasion Assay

The invasion of VD3-GNP-treated (24 h) MDA-MB-231 and MCF-7 cells are 48.38% and 61.37% lower than the untreated controls (*p* < 0.001, *p* < 0.001); see Figure 6A–F. Figure 6A,D show representative micrographs of control cells (MDA-MB-231 and MCF-7), while 6B,E show VD3-GNP-treated cells (MDA-MB-231 and MCF-7). The individual or clusters of Giemsa-stained cells are black in the gray background of the cell insert. The cell migration for the MDA-MB-231 and MCF-7 controls (left panel: Figure 6C) are 48.38% and 61.37% higher than the respective VD3-GNP-treated groups (right panel: Figure 6C,E).

#### 2.2.3. Wound Healing Assay

The breast cancer cell lines (MCF-7 and MDA-MB-231) were used for the cell scratch-based wound healing assay in the presence of the VD3-GNP treatment. This treatment for 24 h causes a marked reduction in the wound healing capacity in TNBC (MDA-MB-231 cells having 38.2% less wound closure) compared to the control cells (*p* < 0.001) (Figure 7A–C), as well as for ER+ MCF-7 cells (27.7% less wound closure) compared to the control cells (*p* < 0.001); see Figure 7D–F.

#### 2.2.4. Cancer Cell Static Adhesion Assay

In the cancer cell static adhesion assay, 24 h of VD3-GNP treatment of MDA-MB-231 and MCF-7 cells causes a marked reduction in the cancer cell static adhesion capacity (48.8% and 92.3%) compared to respective controls (both cases *p* < 0.001); see Figure 7G,H.

#### 2.2.5. Western Blots of PI3K/AKT/mTOR

The VD3-GNP treatment (24 h) of MDA-MB-231 cells lowered protein expression levels, as follows: PI3K by 63.14% (*p* < 0.001); AKT by 75% (ns); p-AKT (SER 473) by 73.73% (*p* < 0.05); mTOR by 65.34% (ns); and ETV7 by 37% (ns), compared to the untreated controls as obtained from Western blots; see Figure 8A,B. In the case of MCF-7 cells, VD3-GNPs downregulated the PI3K levels by 78.3% (*p* < 0.05); AKT levels by 47% (ns); p-AKT (SER 473) (ns) levels by 66.3% (ns), mTOR levels by 214.4% (ns), and ETV7 levels by109.4% (*p* < 0.01), compared to the untreated controls; see Figure 8C,D. The p-AKT/AKT levels (MDA-MB-231 cells) are 70% and 75% for the control and VD3-GNP-treated cells, respectively. For MCF-7 cells, the control and VD3-GNP treatments upregulated the p-AKT/AKT levels by 400% and 30%, compared to the controls. These indicate that the VD3-GNP treatment downregulates the phosphorylation of the AKT in MCF-7 breast cancer cells.

#### 2.2.6. Hippo Pathway Western Blots

The expression levels for the key proteins of the Hippo pathway in MDA-MB-231 and MCF-7 cells were quantified after VD3-GNP treatment (24 h). VD3-GNPs lowered the different proteins levels of, e.g., p-YAP in MDA-MB-231 cells by 59.6% (*p* < 0.05); YAP by 48.93% (*p* < 0.05); and TAZ by 68.12% (*p* < 0.05), compared to the untreated controls (Figure 9A,B). In the case of MCF-7, VD3-GNP treatment for 24 h lowered p-YAP by 39.11% (*p* < 0.05); YAP by 22.86% (ns); and TAZ by 29.53% (*p* < 0.05) (Figure 9C,D), compared to the untreated controls. For control and VD3-GNP-treated MDA-MB-231 cells, p-YAP/YAP are altered by 120% and 60% and in MCF-7 cells are altered by 100% and 80%.

### 2.3. PPI Network (STRING Software 12.0) for the Pathway Proteins (PI3K/AKT/MTOR and Hippo)

Proteins’ functional interactions form the backbone of the cellular machinery and knowledge about protein–protein interactions (PPIs) significantly clarify the molecular mechanisms of the biochemical processes and cellular pathways related to breast cancer [56,57]. Identification of these interactions promises a better understanding of metastasis mechanisms and treatments to disrupt such interactions. We used the STRING database to construct a PPI network for AKT (gene id: AKT1), PI3K (gene id: PIK3CA), MTOR (gene id: MTOR), ETV7 (gene id: ETV7), YAP (gene id: YAP1), and TAZ (Gene ID: TAFAZZIN), to identify their functional partners. A network for 10 possible predicted functional interacting partners is shown in Figure 10. These networks reveal the importance of the therapeutic modulation of PI3K/AKT/MTOR, ETV7, and YAP/TAZ, as they alter several oncogenes and cellular pathways and that can cause several diseases (including breast cancer); details given below.

The PPI network of AKT reveals its important functional partners of different pathway regulators such as the *FOXO3* gene, aging pathway controller; the *NOS3* gene, key regulator of nitric oxide production; and MTOR and PI3K, cell signaling pathway controllers. Among the important functional partners of PI3K are the tumor suppressor gene PTEN, KRAS-RAS signaling pathways; the *IRS1* gene, a key controller of glucose metabolism; and the oncogene *HRAS.* MTOR’s PPI pathway consists of other signaling pathway proteins, e.g., MAPKAP1—MAPK associated protein 1—which acts on the MAPK signaling pathway, as well as the autophagy pathway protein ULK1. For ETV7, the ETS transcription factor’s functional partners are various pathway regulators and some of them are immune regulator proteins like KIR3DL1, KIR3DL2—Killer Cell Immunoglobulin-Like Receptor—and ETV6, which controls hematopoiesis and more cellular pathways.

The Hippo cascade’s key regulatory protein YAP’s PPI network functional partners are ACAD9—a mitochondrial enzyme; ZNF552; ZNF496; and ZNF586—a transcription repressor/activator of DNA or RNA polymerases [58]. In the case of TAZ, the PPI functional partners are inner mitochondrial membrane proteins, which act as mitochondrial translocases like TIM17A, TIM17B, TIM23, and TAMM 41, as well as several solute carriers that act on mitochondria, e.g., SLC25A4 and SLC25A6.

## 3. Discussion

Vitamin D3 gold nanoparticles in breast cancer cells are shown to downregulate migration and invasion. Gold nanoparticles have been used for breast cancer in diverse applications—the evaluation of pathology specimens and non-invasive in vivo imaging, as well as in theragnostic purposes [38,39]. Our newly conjugated gold nanoparticles with a vitamin dosage act by downregulating the novel transcription factor ETV7 and two important cancer signaling pathways. In our earlier studies, higher doses of Vitamin D3 were used (up to 1 mM range) and revealed the downregulation of glycolysis and mTOR, as well as the alteration of EMT pathway proteins (Vimentin, E-cadherin, and N-cadherin) and lowering breast cancer cell viability [59]. Other studies have established an inverse relationship between VD3 and cancer incidence, but high doses of VD3 can cause cytotoxicity in breast cancer cell lines [6,60,61]. GNPs conjugated with VD3 (VD3-GNP) are non-cytotoxic, as revealed using MTT assays (Figure 5). Also, VD3-GNPs are effective at the nM range of VD3 (38 nM (10 mL), as per HPLC studies (Figure 4A,B). The benefit of using nanoparticle-mediated drug delivery is its low dosage, which can avoid cytotoxicity and hypercalcemia, because higher dosages of both VD3 and GNPs can cause cytotoxicity-related cell death, apoptosis, and epigenetic alteration [45,46,47,48,49,50,51,52,53,61,62,63]. Gold, unlike other nanomaterials (e.g., metals like silver and copper, or non-metal nanoparticles like carbon and silica), is non-toxic in low dosage, while, in higher dosages, it causes cytotoxicity in TNBC cells, but has no effect on MCF-7 cells [47,48,49,50,51,52,54,55,62,63]. This low dosage is highly effective in downregulating breast cancer cells’ aggressiveness via the alteration of cellular invasion, a marked reduction in wound healing, and static adhesion.

Nanoparticle size is important in cellular uptake, intracellular delivery, and therapeutic effectiveness [47,48,49,50,51,54,55]. A prior study used 60 nm diameter VD3 gold nanoparticles to induce osteogenic differentiation [50]. In contrast, our VD3-GNPs’ diameters are in the range of ~14–19 (avg~16 nm) nm, as obtained from TEM; the diameter was ~35 nm, as obtained using DLS, while the GNP diameter was ~26 nm using DLS; see Table 1. Our VD3-GNPs are highly effective—even with a low concentration of VD3–potentially enabling a more efficient cellular uptake and anti-cancer action [46,47,48,49,50,51]. Our TEM images revealed a VD3 corona outside the GNP surface (2E,F), while protein coronas outside the GNP surface have also been reported in the literature [52].

In our nano preparation, PDI values of GNPs and VD3-GNP were >1 (see Table 1) and the nanoparticles tended to aggregate, as evidenced by some larger particles imaged using TEM (Figure 2J); hence, we sonicated the preparation for 30 min before use.

We also checked the VD3-GNP conjugation using FT-IR (Figure 3B), where the 2400–2000 cm^−1^ peak corresponds to the thiol bond (-SH) on the GNP surface, indicating the conjugation of VD3 with GNPs, which is also revealed from the TEM images (Figure 2E,F). The *ζ*-potential (ZP), the electrostatic potential at the boundary of the colloidal particles, is an important parameter for different applications, including the characterization of biomedical polymers, the electrokinetic transport of particles or blood cells, and the membrane efficiency of biomaterials [49,52,53]. Our VD3-GNP formulation’s ZP (−24.4 ± 1.8 mV) indicates that it is stable for cellular studies.

Surprisingly, the VD3-GNP treatment is found to downregulate ETV7, the PI3K/mTOR/AKT cascade, along with the HIPPO pathway’s key proteins YAP and TAZ, which indicates its importance in controlling breast cancer aggressiveness (Figure 8 and Figure 9) [35,36,39,40,41,42,43]. A recent report shows increased ETV7 expression [15,16,17,18,19,20,21,22] in all types of breast cancer, compared to normal breast tissue, along with other cancers, while ETV7 expression is correlated with tumor aggressiveness and stemness [14,15,16,17,18,19,20], though its therapeutic downregulation has not yet been seen. ETV7 also acts as an mTORC3 regulator [19]. Downregulating ETV7 with VD3-GNPs can potentially control breast cancer stemness, inflammation, and chemoresistance. Recent studies suggest that ETV7 controls breast cancer inflammation by controlling the TNF-α axis and cross-talking with STAT3—an inflammation regulator—suggesting that VD3-GNPs therapeutically alter these pathway markers via ETV7 [14,15,16,17,18,19,20], which is a challenge for future studies. Moreover, VD3 downregulates mTOR by stimulating DDIT4 and REDD1, as well as cell cycle arrest (at the G1 stage) and senescence [36,43]. A similar mechanism may downregulate PI3K, AKT, and ETV7 levels and thereby reduce breast cancer cells’ invasiveness via the proposed VD3-GNP (24 h) treatment (Figure 8).

We found that VD3-GNP treatment significantly lowers the protein expression levels of YAP/TAZ cascades, as well as the p-YAP (Ser 127) levels in breast cancer cells (MCF-7 and MDA-MB-231 cell lines, Figure 9A–D), which highlights the possible modulation of cross-talk between the Hippo and PI3K/mTOR/AKT pathways by VD3, even at a very low dosage (nM range) [64]. The Hippo pathway and its two key executors, YAP/TAZ, are important upstream signaling pathway regulators for cell proliferation, wound healing, survival, invasiveness, apoptosis, colony formation, and DNA repair in cancer (including breast) [21,22,23,24,25,26,27,28,29]. Major YAP/TAZ target genes (e.g., *CTGF*, *CYR61*, *MYC*, *AXL*, *BIRC5*, and *CCND1*) include molecules that actively control cell growth, proliferation, and migration [21,22,23,24,25,26,27,28,29]. The phosphorylation of YAP/TAZ results in the inhibition of cell proliferation, as we found the lowering of cell migration, adhesion, and invasion after VD3-GNP treatment. Phosphorylation at Ser127 on YAP leads to cytosolic retention, through recognition and binding by the acidic adapter protein 14-3-3, essentially restricting YAP from entering the nucleus and preventing its transcriptional activity; our VD3-GNP treatment lowers it, thus reducing the cytosolic fraction; therefore, future nuclear studies will reveal more about the Hippo cascade [65]. Further studies are needed to understand the role of amino acid transporters and the role of transcription factors (and miRNAs, Figure 1), to understand the cross-talk mechanism and the mode of action of VD3-GNPs.

The VD3-GNPs’ action on key pathway proteins in PPI networks (Figure 10, Appendix A higher-resolution images) shows that it can modulate several cascades; therefore, it can be a future promising therapy in breast cancer patients. We further investigated the TCGA database obtained from UALCAN software (shown in Appendix A), which indicates the upregulation of AKT, mTOR, ETV7, and TAZ in tumor patients. The downregulation of PI3K, YAP, p-PI3K, and p-YAP is important for tumor regression [8,9,10,11,12,13,14,15,16,17,18,19,20,21,22,23,24,25,26,27,28,29]. The VD3-GNP treatment of breast cancer cells downregulated all these proteins and, therefore, the cancer cell metastasis mechanisms, which will be useful for cancer treatment [8,9,10,11,12,13,14,15,16,17,18,19,20,21,22,23,24,25,26,27,28,29].

Up to now, the KEGG pathway diagram of Vitamin D’s mode of action establishes that, upon binding with the Vitamin D receptor (VDR), Vitamin D acts on PI3K/AKT/mTOR regulation (Appendix A). This also demonstrates that our observed VD3-GNP-mediated downregulation of YAP, TAZ, and ETV7 is newly found in breast cancer cells.

Our results establish the beneficial role of VD3-GNPs, which suggests their potential action as non-toxic cancer prevention agents and as a treatment for breast cancer.

## 4. Materials and Methods

### 4.1. Materials

Gold (III) chloride hydrate (99.999% purity), sodium citrate, sodium chloride, tris-salt, Tween, MTT, and Giemsa solution were purchased from Sigma Aldrich (Burlington, MA, USA). Dulbecco’s modified eagle medium (DMEM), fetal bovine serum (FBS), penicillin and streptomycin (PS), 0.5% trypsin, and Dulbecco’s phosphate-buffered saline (DPBS) were purchased from GIBCO (Grand Island, NY, USA). VD3-PEG (Vitamin D3 polyethylene glycol) was purchased from NANOCS (New York, NY, USA) and the 0.22 µm filter cartridges were bought from Millipore (Carrigtwohill, Ireland). MDA-MB-231 cells were purchased from ATCC (American Tissue Culture Collection, Manassas, VA, USA). Tissue culture plate inserts were purchased from VWR Inc. (Radnor, PA, USA) and collagen solution was obtained from BD Bioscience Inc. (Franklin Lakes, NJ, USA).

### 4.2. Synthesis of GNPs and VD3-GNPs

Aqueous dispersions of GNPs were synthesized via the citrate reduction of HAuCl_4_, using a modified Turkovich method [66]. To synthesize ~15 nm GNPs, 0.02% HAuCl_4_ solution (50 mL) was refluxed and then a 2% sodium citrate solution (5 mL) was slowly added dropwise into the flask, while the solution was stirred with a magnetic bar at 200 rpm. Each solution was dissolved in RO (Reverse Osmosis) water. The color of the solution changed from yellow to deep purple. Then, the solution was stirred overnight and was then filtered using a 0.22 µm filter (Millipore, Carrigtwohill, Ireland). After synthesis, the GNPs were stored in RO water at 4 °C. To conjugate vitamin D with GNPs, VD3-PEG was dissolved at a concentration of 5 mg/mL in DMSO and stored at −20 °C. It was diluted 100 times in RO water (to reduce the DMSO concentration 100-fold). VD3-GNPs were fabricated (1:1 volume ratio of GNPs (diluted 100 times) and VD3-PEG solution (50 µg/mL)), while the mixture was vigorously stirred at room temperature for 24 h. The mixture was filtered with a 0.22 µm filter, before using it for assays and it was then sonicated for 30 min. A schematic of the GNP and VD3 synthesis process is shown in Figure 11A,C and the Vitamin D3 structure is shown in Figure 11B.

### 4.3. VD3-GNPs Characterization Using Biophysical Methods

#### 4.3.1. TEM Microscopy

The synthesized GNPs and VD3-GNPs were scanned with FE-TEM (Hitachi H-9500, H-7650, H-8100 TEM at 300 kV, Hitachi High-Tech America, Inc., Pleasanton, CA, USA), to identify the morphology of the VD3-GNPs. The GNPs and VD3-GNP solutions were dropped onto a 200-mesh copper grid coated with Formvar/carbon (Ted Pella Inc., Redding, CA, USA) and were dried under vacuum at room temperature. Any remaining solutions were removed with filter paper before TEM analysis. After complete drying, the GNPs and VD3-GNPs were imaged at 100 kV. For VD3-GNPs, n = 111 and, for GNPs, n = 202 particles were quantitated using AMT 600 software, (Hitachi) and 70,000–120,000× magnification was used.

#### 4.3.2. UV–Visible (UV–Vis) Spectroscopy

The optical properties of the gold nanoparticle solutions were evaluated in water and buffer with a nanodrop spectrophotometer/fluorometer (DeNovix, DS11 FX, Wilmington, DE, USA), using the nanodrop function at 250–650 nm.

#### 4.3.3. FT-IR Spectrum Measurement

Infrared absorption spectra were recorded on a Bruker Optics Vertex 70 (Bruker Scientific, LLC, Billerica, MA, USA) Fourier transform infrared (FTIR) spectrometer, equipped with a Bruker Optics Platinum attenuated total reflection (ATR) accessory (diamond ATR crystal). We applied the attenuated total reflection setup in the diamond ATR prism. The ATR method was applied using an IR source—a KBr beam-splitter. The background spectrum was recorded (buffer and ethanol) using an NCD-coated ATR prism without VD3-GNPs or VD3. The dispersion of 100 µL of VD3-GNPs and GNPs in 1 mL PBS were ultrasonicated for about 1 h before use, forming an optically homogeneous dispersion. The nanoparticles were applied to the ATR prism by drop-casting 10 µL of dispersion solution and the measurement was taken. During the IR measurements, the VD3-GNPs were in the N_2_-purged chamber of the FTIR spectrometer. The advanced ATR correction was applied to the measured spectra. The buffer and ethanol backgrounds were collected and subtracted before calculation. The ATR prism was cleaned with wet cotton buds, followed by an additional control spectral measurement to show that all absorbance peaks vanished from the spectrum.

#### 4.3.4. Particle Size Measurement Using DLS

GNPs’ and VD3-GNPs’ diameters and polydispersity indices were measured using DLS (Microtrac Zetatrac particle size; Colloid Metrix GmbH, Meerbush, Germany). The GNP and VD3-GNP solutions were diluted (1:10) from the stock for the measurement.

The polydispersity index is defined as follows:

PDI = M_w_/M_n_,

M_w_ = Weight-average molecular weight

M_n_ = Number-average molecular weight.

This number is calculated by the Microtrac Zetatrac and is widely used (see, e.g., [67,68]). A PDI value of 1 indicates that particles are monodisperse, whereas a PDI value larger than 1 suggests a polydisperse sample. We expect that synthetic polymers, like PEG, have a PDI value > 1. In our experiments, we found that PDI ranges up to 1.6, which can result from multi-particle aggregation (further addressed in the Section 3).

#### 4.3.5. Zeta Potential

The effective surface charges on the gold nanoparticles were measured using a zeta potential analyzer (Particle Metrix Stabino, Verder Scientific GmbH & Co, Haan, Germany) [52]. Zeta potential measurements were made in 3 mL water, containing 10 mL of nanoparticle solution. Data were obtained using a monomodal acquisition and were fit according to the Smoluchowski theory.

#### 4.3.6. Quantitation of the Vitamin D3 in VD3-GNPs Using HPLC

A Dionex Ultimate 3000 HPLC system equipped with an autosampler, column temperature control chamber, and PDA detector for chromatographic analysis was used for vitamin D3 quantitation. The analysis was carried out using a Thermo Scientific Acclaim 120 C18 column, with dimensions of 120 Å 3 × 150 mm and a particle size of 3.0 µm. The mobile phase consisted of 100% acetonitrile (HPLC grade) at 40 °C, with a flow rate of 0.425 mL/min. The detection was performed at 265 nm. Samples were injected using a six-port injection valve equipped with a 20 µL loop. The instrument was controlled and data acquisition was performed using Chromeleon 7.2.10 software. The VD3-GNP sample was prepared as mentioned earlier and 100 µL was evaporated using a speed vac (Integrated Speed vac, Thermofiher Scientific, Waltham, MA, USA) and was then resuspended in 5 µL acetonitrile and 995 mL acetonitrile (HPLC grade), for the final HPLC analysis. Standard calcitriol 10 mM (Vitamin D3) was also prepared in a similar manner (5 µL standard +995 µL acetonitrile) and was used for HPLC analysis [69]. The following equation was used to calculate the concentration of VD3 in the VD3-GNPs:

Concentration of sample = area of sample/area of standard × concentration of standard.

### 4.4. Cell-Based Studies

#### 4.4.1. Breast Cancer Cell Lines’ Maintenance and MTT Assay after VD3-GNP Treatment

MDA-MB-231 and MCF-7 cells were purchased from ATCC and were maintained in DMEM, containing 10% heat-inactivated FBS, 100 U/mL penicillin, and 100 µg/mL streptomycin. Cells were split to 100,000–250,000 cells/mL twice per week using Trypsin-EDTA (2–3 min for volume 1–1.5 mL until the cell dislodge), by pelleting the cells at 1000× *g* for 5 min.

The cytotoxicity evaluation of VD3-GNPs was performed using the MTT assay [64,70]. Approximately 1 × 10^5^ cells mL^−1^ (MDA-MB-231 and MCF-7) in their exponential growth phase were seeded in flat-bottomed 96-well polystyrene plates and were incubated for 24 h at 37 °C in a 5% CO_2_ incubator. In the medium, a series of VD3-GNP dilutions (10 µg/mL with 100 µL per well, 1:10, 1:100, and 1:1000 from 10 µg/mL stock) were added to the plate in triplicate. After 24 h of incubation, 50 µL of MTT reagent was added to each well and further incubated at 37 °C in the dark for 2 h. The formazan crystals that formed after 2 h of incubation in each well were dissolved in 150 µL of DMSO and each well was read with a spectrophotometer/fluorometer (DeNovix, DS11 FX, Wilmington, DE, USA), using the nanodrop function at 570 nm, with background correction at 630 nm. Wells with complete medium and nanoparticles, MTT reagent, and without cells were used as blanks.

#### 4.4.2. Cancer Cell Invasion

Serum-induced cell invasion was performed at 37 °C for 24 h using a 24-well trans well insert coated with collagen solution overnight (BD Biosciences Inc). A total of 1 × 10^4^ MDA-MB-231 and MCF-7 cells were suspended in 250 µL of serum-free DMEM medium and were seeded into the upper chamber of each insert (24-well insert; pore size 8 µm; BD Biosciences). Then, 600 µL of DMEM, containing 10% FBS, was added to a 24-well plate, for the control sets. In the VD3-GNP treatment group, 500 µL of DMEM, containing 10% FBS+ 100 µL of VD3-GNP solution, was added and incubated for 24 h. After 24 h, the upper surface of the insert was wiped gently with a cotton swab to remove non-migrating cells. Cells that migrated and invaded through the membrane were stained with Giemsa solution (0.5% crystal violet in 25% methanol/DPBS) for 20 min, washed six times with DPBS, and photographed with an inverted microscope with a camera (Nikon Eclipse Ti-E, Melville, NY, USA) in ten random fields at 10× *g* magnification. Cells that migrated to the lower side of the filter were manually counted from the images. The average number of migrated cells was obtained from ten randomly selected images for each condition and was compared statistically. Three independent replicates were performed.

#### 4.4.3. Wound Healing Assay

MDA-MB-231 and MCF-7 cells (2.5 × 10^5^ cells/well) were seeded in 24-well plates, to grow in a monolayer for 24 h. Then, a sterile 200 µL pipette tip was held vertically to scratch across each well. The detached cells were removed by washing with 500 µL PBS and shaking manually for 2–3 min. In total, 500 µL of fresh medium with or without VD3-GNPs was added afterward and was incubated for another 48 h. The breast cancer cell migration was assessed from images taken using a Nikon Eclipse microscope at 10× *g* magnification. The wound width was measured by taking 10 independent width (nm) measurements of each image using ImageJ and the average value was obtained. The control was compared with VD3-GNPs. The percentage of open wound width was plotted, along with experimental group, after 24 h of study. Data are presented as mean ± SEM. Three replicates were included in the analysis. A total of 22 individual images were quantitated and an unpaired Student *t*-test was performed. Significance was considered at *p*  <  0.05.

#### 4.4.4. Cancer Cell Static Adhesion Assay

96-well culture plates were coated with collagen I (1 mg/mL) overnight. MDA-MB-231 and MCF-7 cells were subsequently harvested. Then, 1 × 10^4^ cells were plated onto pre-coated wells as control and were treated with VD3-GNPs (10 mg/mL). The number of adhered cells was determined using the protocol of Rusciano et al. [62], with modification. Specifically, cells were left to adhere for 1 h and the wells were then washed three times with PBS and were subsequently fixed with 70% alcohol for 30 min. Adherent cells were stained with 0.1% crystal violet for 15 min and were washed with water. Stained cells were solubilized in 5% Triton X100 overnight. An ELISA plate reader quantitated absorbance at a wavelength of 570 nm. Data were expressed as the mean of 8 wells ± the SEM.

#### 4.4.5. Protein Extraction for Western Blotting

A total of 2.5 × 10^7^ cells were plated in six well plates and were grown for 48 h until confluent. Then, VD3-GNPs (10 µg/mL) were added and all cells were grown for another 24 h. After that, cells were washed with PBS and lysed using Pierce^®^ IP Lysis Buffer (ThermoFisher Scientific, Waltham, MA, USA), as per manufacturer instructions, before being centrifuged at 12,000× *g* at 4 °C for 10 min. After the spin, the supernatant was collected in fresh tubes and the protein concentration was determined using a spectrophotometer/fluorometer (DeNovix, DS11 FX, Wilmington, DE, USA), with the nanodrop function at 280 nm and using BSA standard 10 mg/mL.

#### 4.4.6. Western Blots

In total, 60 µg of proteins were diluted 1:1 in Laemmli sample buffer (Bio-Rad, Hercules, CA, USA, 161-0737) with β-mercaptoethanol (Bio-Rad, Hercules, CA, USA, 1610710), placed in 4–20% gradient acrylamide gels (Bio-Rad, 456-8094), and separated using electrophoresis. Samples from the gels were transferred to PVDF membranes (Bio-Rad, 1704156) using the Trans-Blot Turbo system (Bio-Rad, 1704150). PVDF membranes were stained with Ponceau S (Sigma, P3504) solution for 5 min, to verify protein transfer and were washed three times with TBS-T (Tris Buffer Saline (100 mM Tris-HCl pH 7.5, 150 mM NaCl) and 0.1% Tween (Bio-Rad, 170-6531)). PVDF membranes were blocked with 5% BSA in TBS-T at room temperature for 1 h and were subsequently incubated with primary antibodies overnight at 4 °C, followed by incubation with anti-mouse IgG (1:10,000, Jackson-Immuno Research, West Grove, PA, USA, 115035068) in TBS-T for 1 h. Primary antibodies for PI3K (Developmental Studies Hybridoma Bank (DSHB), AFFN-PIK3R2-3D4), mTOR (DSHB, CPTC-MTOR-1), AKT (DSHB, CPTC-AKT1-1), ETV7 (DSHB, PCRP-ETV7-1A1), YAP (DSHB, YAP2), Taz (DSHB, PCRP-ZBTB18-1B2), p-AKT (Ser473), and p-YAP (Ser127), (Cell Signaling Technology, Danvers, MA, USA, 9271) were used (1:100 in 0.5% TBS-T) and β-actin antibody conjugated to peroxidase was used as a loading control (Sigma-Aldrich, A3854, dilution 1:50,000). The blots were developed using a horseradish peroxidase (HRP) chemiluminescent substrate reagent kit (Biorad, Hercules, CA, USA, 1705060) and were visualized using a blot image analyzer system (Bio-Rad, ChemiDocMP, Taunton, MA, USA). Expression levels were quantified using ImageJ software 1.54i. All blots were normalized by the β-actin loading control. Three independent replicates were obtained.

### 4.5. PPI Network (STRING Software 12.0) for the Pathway Proteins (PI3K/AKT/MTOR and Hippo)

#### 4.5.1. Construction of PPI Networks (STRING)

The online data analysis software STRING (Search Tool for Retrieval of Interacting Genes/Proteins) was used to build the protein–protein interaction (PPI) network for AKT (Gene ID: AKT1), PI3K (Gene ID: PIK3CA), mTOR (Gene ID: MTOR), YAP (Gene ID: YAP1), and TAZ (Gene ID: TAFAZZIN), with 10 possible functional partner proteins, using a high confidence level of 0.7 [56,57] and, for ETV7, using a low confidence level of 0.4. In these, PPI networks nodes represent proteins, splice isoforms, or post-translational modifications, i.e., each node represents all the proteins produced by a single, protein-coding gene locus. According to the STRING database, two interacting proteins—both contributing to a specific biological function—act as functional partners. The STRING database constructs a PPI network by collecting, scoring, and integrating all publicly available sources of protein–protein interaction information and complements these with computational predictions, where it aims to achieve a comprehensive and objective global network, including direct (physical), as well as indirect (functional), interactions (protein network details have been added in Appendix A).

#### 4.5.2. Network Details

In our current protein network model, the network nodes representing proteins, splice isoforms, or post-translational modification are collapsed, i.e., each node represents all the proteins produced by a single protein-coding gene locus.

Node colored ‘ball shaped nodes’ are proteins in the first shell of interactors; white nodes are in the second shell of interactors. Node content—empty nodes are proteins of unknown 3D structure; filled nodes indicate that the 3D structure is known or predicted. Edges represent protein–protein associations. Edge colors—curated database (cyan edge); experimentally determined (pink edge); predicted interactions—gene neighborhood (green edge); gene fusions (red edge); gene co-occurrence (blue edge). Others—text mining (yellow edge); co-expression (black edge); protein homology (blue edge).

### 4.6. Statistical Analysis

The experimental data were represented by the mean ± standard error of the mean (SEM) of at least three samples. TEM images size measurements were represented as mean ± standard deviation (SD). All experimental data were used for the statistical analyses. The statistical analyses were performed for the comparison of the two groups using the paired student *t*-test and ANOVA and Tukey’s post hoc analysis, for comparing more than two samples when the data were normally distributed, or the signed rank sum test when the data were not normally distributed. A *p*-value < 0.05 was considered statistically significant.

## 5. Conclusions

We have synthesized VD3-GNPs and quantitated their biophysical properties using different techniques, e.g., TEM, DLS, zeta potential, UV–Vis spectroscopy, FT-IR, and HPLC. A non-toxic concentration of VD3-GNPs was used for all cell-based experiments (in MDA-MB-231 and MCF-7 cells, based on the MTT assay), as the toxicity of VD3 (at higher dosage) causes several health problems and both VD3 and GNPs (at higher concentrations) are cytotoxic to cells.

We have established that, at nanomolar doses, VD3-GNP treatment (24 h) mitigates breast cancer (MDA-MB-231 and MCF-7) cells’ aggression, by reducing invasion, migration, and adhesion, along with downregulating important proteins in cell signaling pathways, e.g., PI3K/mTOR/AKT, including the transcription factor ETV7 and Hippo cascades (quantitated using Western blots). Furthermore, we have established the importance of pathway protein upregulation in human tumors (breast cancer samples), using the UALCAN software and protein–protein interactions (PPI network) between the key proteins in these two cascades, using STRING software to highlight the importance of the cell signaling pathways.

Hence, VD3-GNPs are a promising candidate for clinical therapy. Our future studies will target the gene expression profiles and additional regulatory proteins in the PI3K, Hippo, and other pathways (such as autophagy, apoptosis, etc.).

## 6. Limitations

In the current study we did not use large, potentially cytotoxic, concentrations of VD3; so, vitamin D toxicity was not addressed. Resource limitations prevented the inclusion of additional cell lines and other time points, as well as animal models. Two breast cancer cell lines are sufficient to establish our initial hypotheses [70,71,72,73,74]. Also, the action of VD3-GNPs in TNBC and ER (+) patients was not explored; future clinical studies will address this.

## Figures and Tables

**Figure 1 ijms-25-05348-f001:**
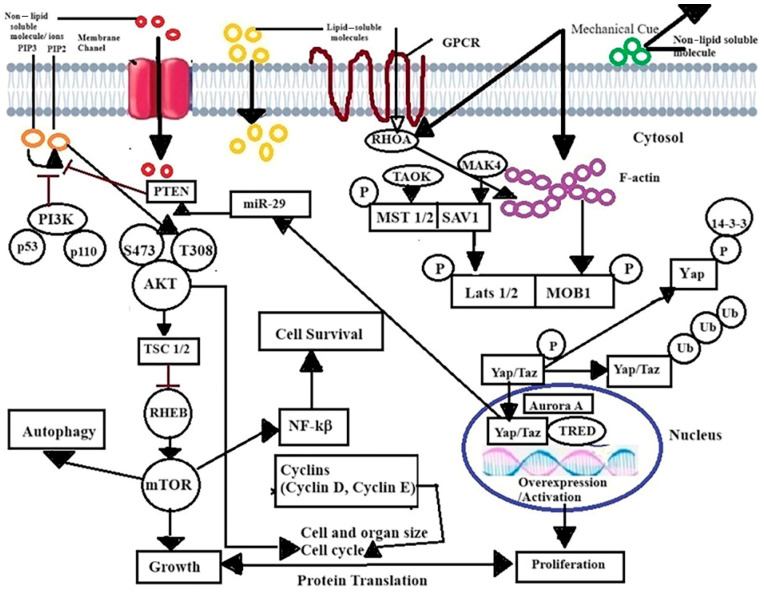
Schematic of the PI3K/AKT/mTOR and Hippo pathways and their roles in various cellular mechanisms (growth, proliferation, etc.).

**Figure 2 ijms-25-05348-f002:**
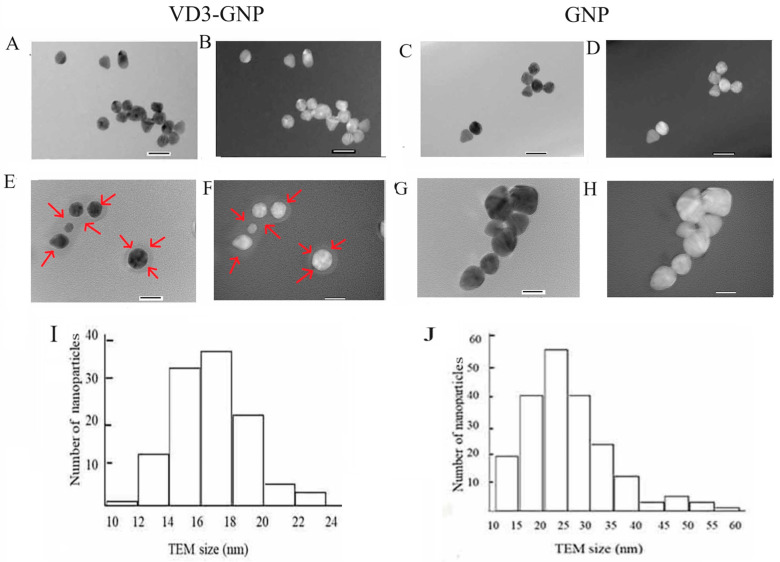
(**A**) TEM image of the VD3-GNPs; (**B**) TEM image (inverted) of the VD3-GNPs; (**C**) TEM image of the GNPs; (**D**) TEM image (inverted) of the GNPs at magnification 300,000×; (**E**) TEM image of VD3-GNPs with VD3 corona (magnification 200,000× and print magnification 1080,000×@7in); (**F**) TEM image (inverted) of VD3-GNPs with VD3 corona (magnification 200,000× and print magnification 1080,000×@7in). Red arrows are showing the VD3-GNP corona; (**G**) TEM image of GNPs without any corona (magnification 200,000× and print magnification 1080,000×@7in); (**H**) TEM image (inverted) of GNPs. Scale bar represents 20 nm; (**I**) TEM image size distribution diagram of VD3-GNPs (n = 111); (**J**) TEM image size distribution diagram of GNPs (n = 202).

**Figure 3 ijms-25-05348-f003:**
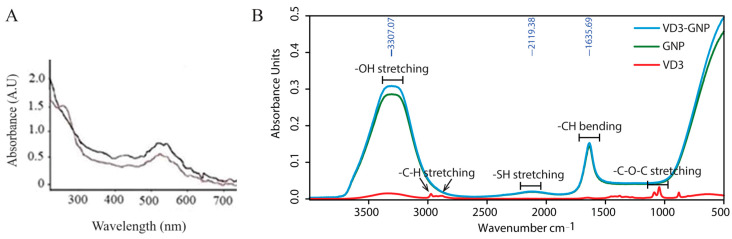
(**A**) UV–Vis VD3-GNPs (black) and GNPs (brown) spectra; (**B**) FT-IR spectra of the VD3-GNPs (blue), GNPs (green), and VD3 (red).

**Figure 4 ijms-25-05348-f004:**
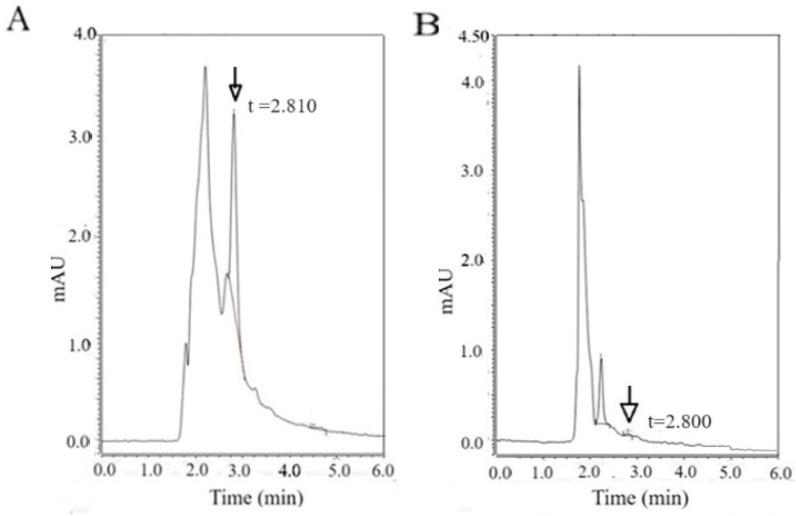
(**A**) HPLC elution profile of Vitamin D3 (concentration—10 mM) and (**B**) VD3-GNPs. The VD3 retention peak is at 2.8 min, marked by an arrow.

**Figure 5 ijms-25-05348-f005:**
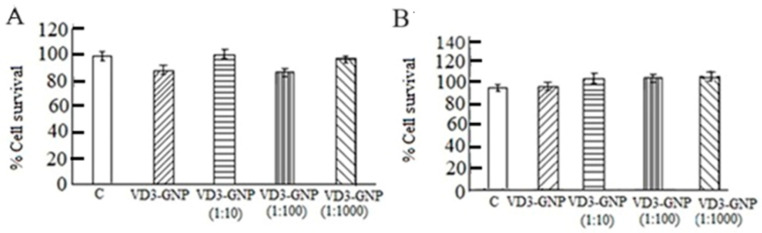
(**A**) % of MCF-7 cells survival using the MTT assay after 24 h of VD3-GNP treatment (ns); (**B**) % of MDA-MB-231 cell survival using the MTT assay after 24 h of VD3-GNP treatment (ns).

**Figure 6 ijms-25-05348-f006:**
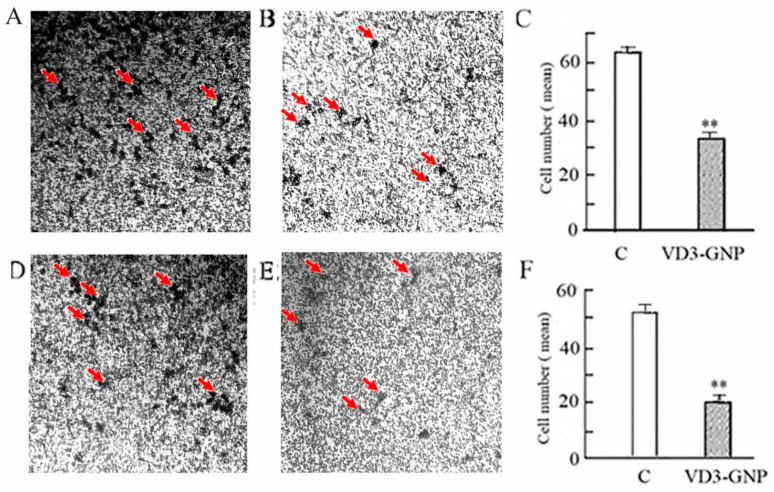
Cell invasion assay. (**A**,**B**) Representative micrographs of control (MDA-MB-231) cells and VD3-GNP-treated (MDA-MB-231) cells, after Giemsa stain. Images were contrast-enhanced by 118%. (**C**) Quantification of MDA-MB-231 cell invasion assay (** *p* < 0.001). (**D**,**E**) Representative micrographs of control (MCF-7) cells and VD3-GNP-treated (MCF-7) cells, after Giemsa stain. Images were enhanced by 133%. (**F**) Quantification of MCF-7 cell invasion assay (** *p* < 0.001). Three independent replicates were performed for each cell line. Invaded cells are indicated by an arrow in the micrographs.

**Figure 7 ijms-25-05348-f007:**
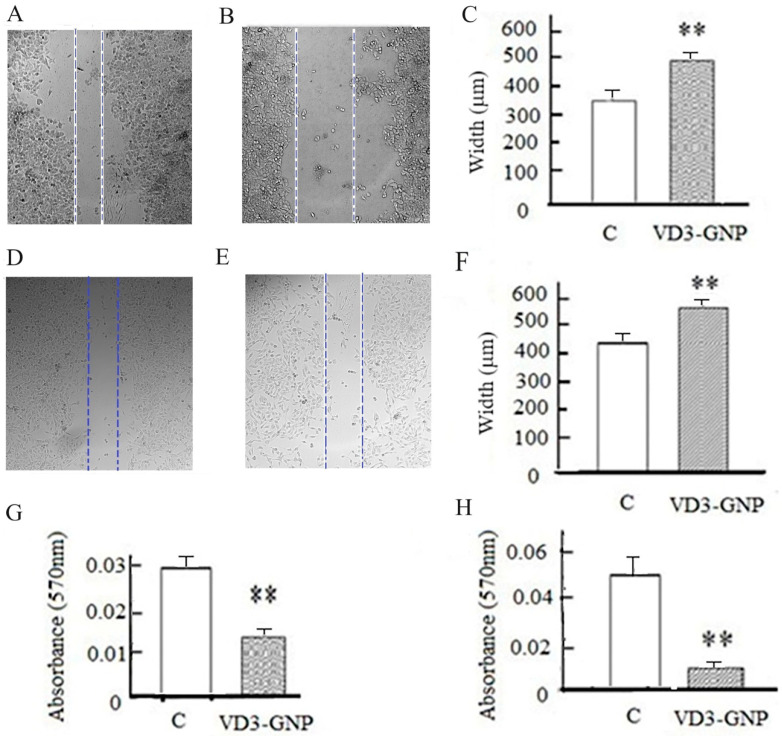
Cancer cell wound healing assays. (**A**) Representative micrograph of control (MDA-MB-231) cells; (**B**) representative micrograph of VD3-GNP-treated (MDA-MB-231) cells after 24 h of treatment; (**C**) average wound width (µm) mean ± SEM of MDA-MB-231 cells (** *p* < 0.001); (**D**) representative micrograph of control (MCF-7) cells; (**E**) VD3-GNP-treated (MCF-7) cells after 24 h of treatment; (**F**) wound width (µm) of MCF-7 cell wound healing assay (** *p* < 0.001); (**G**) static cell adhesion assay of MDA-MB-231 cells (** *p* < 0.001); and (**H**) static cell adhesion assay of MCF-7 cells (** *p* < 0.001). Images in (**A**,**B**) were contrast-enhanced by 125% and (**D**,**E**) were contrast-enhanced by 180%.

**Figure 8 ijms-25-05348-f008:**
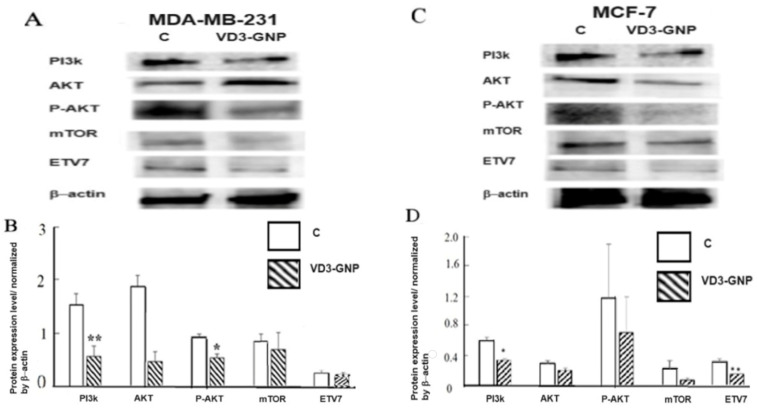
(**A**) Representative Western blots of PI3K, AKT, p-AKT(Ser473), mTOR, and ETV7 in control (**C**) and VD3-GNP-treated MDA-MB-231 cells. * *p* < 0.05. (**B**) Calculation of the protein expression (band density) in Western blot analysis (n = 3) in MDA-MB-231 cells. Protein levels of PI3K, AKT, p-AKT (Ser473), mTOR, and ETV7 were normalized by β-actin, * *p* < 0.05, ** *p* < 0.001. (**C**) representative Western blots of PI3K, AKT, p-AKT (Ser473), mTOR, and ETV7 in control (**C**), and VD3-GNP-treated MCF-7 cells. β-actin was used as a loading control. (**D**) The protein expression (band density) in Western blot analysis (n = 3) for protein levels in MCF-7 cells was normalized by β-actin, * *p* < 0.05, ** *p* < 0.001.

**Figure 9 ijms-25-05348-f009:**
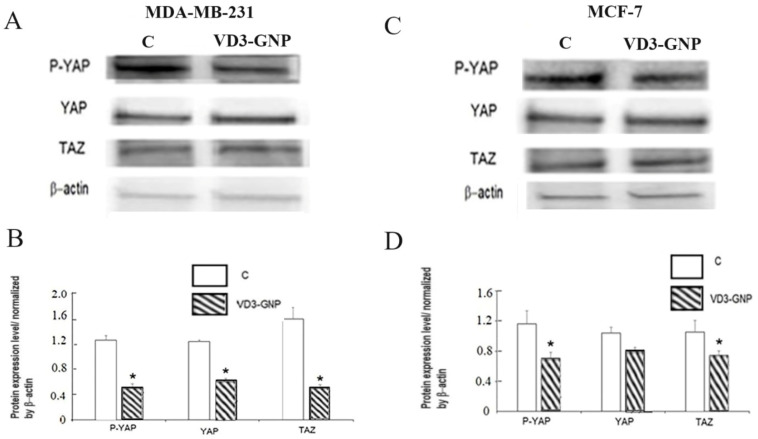
(**A**) Representative Western blots of p-YAP, YAP, and TAZ in control (**C**) and VD3-GNP-treated MDA-MB-231. (**B**) Protein expression (band density) from Western blot analysis (n = 3) for protein levels of p-YAP, YAP, and TAZ, normalized by β-actin in MDA-MB-231cells, * *p* < 0.05. (**C**) Representative Western blots of p-YAP, YAP, and TAZ in control (**C**) and VD3-GNP-treated MCF-7. β-actin was used as a loading control. (**D**) Protein expression (band density) from Western blot analysis (n = 3) for protein levels of p-YAP, YAP, and TAZ, normalized by β-actin in MCF-7, * *p* < 0.05.

**Figure 10 ijms-25-05348-f010:**
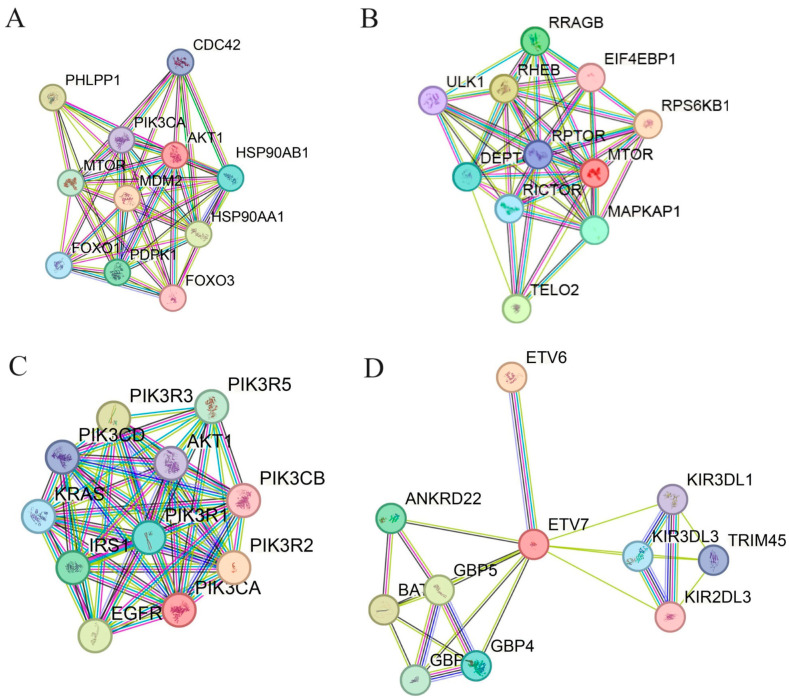
PPIs (using the STRING database) for the PI3K/mTOR/AKT (Gene IDs: PIK3C, MTOR, AKT1, ETV7) and Hippo pathways (Gene IDs: Yap1, TAZ), with 10 functional proteins. (**A**) AKT1; (**B**) MTOR; (**C**) PI3K; (**D**) ETV7; (**E**) YAP; (**F**) TAZ.

**Figure 11 ijms-25-05348-f011:**
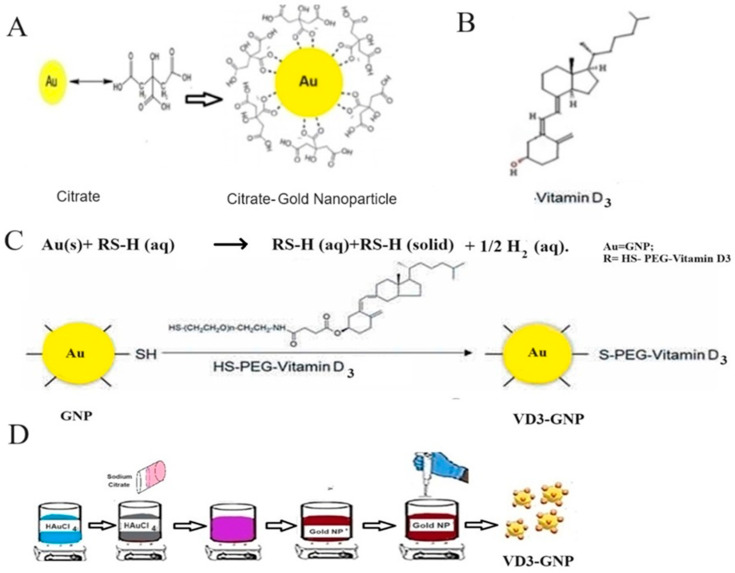
(**A**) Schematic of gold nanoparticle formation; (**B**) vitamin D3 (Cholecalciferol, C_27_H_44_O) obtained from PubChem; (**C**) the VD3-PEG and GNP conjugation reaction during VD3-GNP synthesis; (**D**) schematic of VD3 conjugation with gold nanoparticles.

**Table 1 ijms-25-05348-t001:** Tabular summary of the DLS measurements.

Nanoparticles	DLS (nm)	PDI
VD3-GNP	34.6 ± 0.6	1.622
GNP	27.05 ± 1.8	1.505

## Data Availability

The datasets used and/or analyzed during the current study are available from the corresponding author on reasonable request.

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
