# Peer review of "Mitigation of Breast Cancer Cells’ Invasiveness via Down Regulation of ETV7, Hippo, and PI3K/mTOR Pathways by Vitamin D3 Gold-Nanoparticles"

_ijms, 2024, doi:10.3390/ijms25105348_

Round 1

Reviewer 1 Report

Comments and Suggestions for Authors

In this manuscript, the authors uncovered that the invasiveness of breast cancer cells could be mitigated by Vitamin D3 gold nanoparticles through the downregulation of ETV7, Hippo, and PI3K/mTOR pathways. However, some revisions and additions are needed.

1. It is better to assign peaks to specified bonds and label them in Figure 2.

2. The labeling of Figure 3 is confusing.

3. In some figures (Figure 5, Figure 6, Figure 9, and Figure 10), quality and clarity are low.

4. In Figure 7A, it seems that the expression of AKT protein is increased after treatment, which is not consistent with 7B. Is this because of the comparison to Actin?

Author Response

  1. It is better to assign peaks to specified bonds and label them in Figure 2.
     This is a useful suggestion. We have labeled peaks in our new FT-IR image (Figure 3B) added FT-IR spectrum of GNP and marked the peaks [-OH bond, -SH bond -C-C Bond etc.]

  1. The labeling of Figure 3 is confusing.

     We edited Figure 3 image panes and legends to address this issue, and it is now Figure 3, where we have VD3-GNP (UV-Vis) and FT-IR images (where we incorporated the GNP spectrum and marked the FT-IR peaks).

  1. In some figures (Figure 5, Figure 6, Figure 9, and Figure 10), quality and clarity are low.

     We have modified Figures 5-11 to address this issue by replacing the images with higher-resolution images and increasing the font within the image.

  2. In Figure 7A, it seems that the expression of AKT protein is increased after treatment, which is not consistent with 7B. Is this because of the comparison to Actin?

       The referee’s observation is correct. While the levels of AKT are higher after treatment, the levels of beta-actin are also higher, therefore renormalization of both control and treated cells results in lower relative levels of AKT in the treated group. Also, the data is for two individual cell lines (MDA-MB-231,             MCF-7) so the expression of AKT as well as beta-actin levels are different.

Reviewer 2 Report

Comments and Suggestions for Authors

In this manuscript, the authors synthesized a Vitamin D3 conjugated to gold for the treatment of breast cancer. While the work is good to some extent, there are several concerns regarding the clarity of representations and the logical flow of language throughout the paper. Here are some specific comments:

  1. The introduction lacks organization, and I found it challenging to follow at certain points.
  2. The figures suffer from issues related to layout, color, and isolation.
  3. TEM images of gold nanoparticles are missing.
  4. It's unclear whether the size of the GNP (15 nm) was measured by TEM or DLS.
  5. Figure 3D is not included.
  6. In Figure 5, the background is unclear.
  7. Most experiments, such as the Cell Invasion Assay, do not include GNP or VD3 for comparison with the developed NP.
  8. There is no mention of PEG or polymer until line 170.
  9. Regarding line 459-460, it's unclear if the author measured the GNP and VD3-GNP in the FT-IR spectrum measurement or VD3 and VD3-GNP.
  10. Numerous typographical errors need careful checking to enhance the manuscript's quality.
Comments on the Quality of English Language

There are several concerns regarding the clarity of representations and the logical flow of language throughout the paper.

Author Response

  1. The introduction lacks organization, and I found it challenging to follow at certain points.

    We have shortened the Introduction with better articulation and added subheadings, which should make it clearer. Thank you for the suggestion.

  2. The figures suffer from issues related to layout, color, and isolation.

    We have modified our figures (Figures 2, 3, 5, 7-11, etc.) in our revised manuscript to address this issue like marking peaks in our FT-IR image and adding GNP spectrum (Figure 3B). We have added new TEM image panes in Figure 2 and size distribution curves for TEM, and added bigger microscopic images in Figures 6, 7, etc. This has indeed improved the presentation.
  3. TEM images of gold nanoparticles are missing.

We have performed TEM imaging of both VD3-GNP and GNP. Our current study is not a mere comparative study between VD3-GNP, GNP, and VD3; our primary focus is to study the alteration of cellular signaling pathways, so we didn’t include GNP TEM images earlier. We have now added GNP-TEM in (Figure 2C, D, G, H) as per the reviewer’s request.

  1. It's unclear whether the size of the GNP (15 nm) was measured by TEM or DLS.

Both TEM and DLS were used to quantify the size of VD3-GNP and GNP (Figures 2A-J, Table 1). In our Results and Discussion sections, we clearly mentioned these two measurements (TEM and DLS). In our GNP TEM image due to some big particles (which probably happened due to coagulation of GNPs at high TEM temperature), the mean size is higher (along with higher SD). In the revised manuscript we also added the DLS size (in parentheses) along with TEM to satisfy the reviewer’s concern on page 20. The new sentences are: “In contrast, our VD3-GNPs’ diameters are in the range of ~14-19 (avg ~ 16nm) nm (as obtained from TEM, diameter ~ 35 nm from DLS, while the GNP diameter ~ 26 nm from DLS; see Table 1); our VD3-GNPs are highly effective – even with a low concentration of VD3 –potentially enabling more efficient cellular uptake and anti-cancer action [47-52]. Our TEM images revealed VD3 coronas outside the GNP surface (Figures 2E, F), while   protein coronas outside of the GNP surface have also been reported in the literature [68].”

  1. Figure 3D is not included.

The Figure 3 legend in the revised manuscript has been corrected, and currently, it is Figure 3– thank you for catching this error. It now reads “(A) UV-Vis’s of VD3-GNP (black) and GNP (brown) spectra; (B) The FT-IR spectrum of the VD3-GNP (blue), GNP (green) and VD3 (red).”

  1. In Figure 5, the background is unclear.
    We have modified Figure 5 (enhanced the contrast of the microscopic images) to address the issue raised by this reviewer and now it is Figure 6.

  2. Most experiments, such as the Cell Invasion Assay, do not include GNP or VD3 for comparison with the developed NP.

    Our study is focused on VD3-GNP treatment as mentioned above (response number 3), so we didn’t include GNP data. One recent study by Veeresh et al. 2024 (Reference 65) established VD3’s role in inhibiting breast cancer cells’ viability, along with our previous study (Reference 63). While other studies identified the role of different dosages of gold nanoparticle in breast cancer cells’ radiosensitivity and chemo sensitization, they didn’t investigate cellular signaling pathway alterations (References 65-68) which is the significant finding of our current study when we use VD3-GNP.

    The five new References (65-69), added to address the referee’s comment, are:

65.Veeresh, P. K. M., Basavaraju, C. G., Dallavalasa, S., Anantharaju, P. G., Natraj, S. M., Sukocheva, O. A., Madhunapantula, S. V. Vitamin D3 Inhibits the Viability of Breast Cancer Cells In Vitro and Ehrlich Ascites Carcinomas in Mice by Promoting Apoptosis and Cell Cycle Arrest and by Impeding Tumor Angiogenesis. Cancers (Basel). 2023, 15:4833.

  • Lee, J., Chatterjee, D. K., Lee, M. H., Krishnan, S. Gold nanoparticles in breast cancer treatment: promise and potential pitfalls. Cancer Lett. 2014.347:46-53.
  • Surapaneni, S.K., Bashir, S., Tikoo, K. Gold nanoparticles-induced cytotoxicity in triple negative breast cancer involves different epigenetic alterations depending upon the surface charge. Sci Rep.2018, 8: 12295.
  • Wang, W., Ding, X., Xu, Q., Wang, J., Wang, L., Lou, X. Zeta-potential data reliability of gold nanoparticle biomolecular conjugates and its application in sensitive quantification of surface absorbed protein. Colloids and Surfaces B: Biointerfaces, 2016, 148; 541-548.
  • Khongkow, M., Yata, T., Boonrungsiman, S. et al.Surface modification of gold nanoparticles with neuron-targeted exosome for enhanced blood–brain barrier penetration. Sci Rep, 2019,9, 8278.

       8. There is no mention of PEG or polymer until line 170.

We added PEG in the revised manuscript Introduction (lines 61-62) also in response to this reviewer: “With the goal of developing a non-toxic therapy (see later), we focus on mitigating breast cancer invasion and the related cellular pathways, and we achieve our goal using VD3-GNPs (where VD3-PEG is conjugated with GNP)”

9. Regarding line 459-460, it's unclear if the author measured the GNP and VD3-GNP in the FT-IR spectrum measurement or VD3 and VD3-GNP.

We have now incorporated the FT-IR spectrum (absorbance) of VD3-GNP, GNP, and VD3 and have added them in Figure 3B, where VD3-GNP is shown in blue, GNP is shown in green, and VD3 is shown in red.

10. Numerous typographical errors need careful checking to enhance the manuscript's quality.

We have carefully edited our revised manuscript to address this issue.

Reviewer 3 Report

Comments and Suggestions for Authors

The article titled "Reduction of Breast Cancer Cells' Invasiveness through Suppression of ETV7, Hippo, and PI3K/mTOR Pathways by Vitamin D3 Gold-Nanoparticles" submitted by by Moumita et al provides valuable findings on the potential use of vitamin D3 attached to gold nanoparticles (VD3-GNP) for diminishing the aggressiveness of breast cancer cells. Nevertheless, there exist constraints and aspects that require enhancement, which are crucial for assessing the study's strength and applicability:

1. This study specifically examines two types of breast cancer cell lines, namely MDA-MB-231 (triple-negative) and MCF-7 (ER+). While these models have a strong foundation, incorporating a broader spectrum of breast cancer subtypes would yield a more comprehensive comprehension of the treatment's efficacy across various disease profiles. 

2.  Although the study demonstrates the efficacy of VD3-GNP at nanomolar concentrations, the precise range of dose and duration of therapy that is most suitable for clinical use is yet uncertain. Further dose-response studies could assist in establishing more conclusive therapeutic ranges.

 3. While the study shows a decrease in the activity of important pathways and proteins like ETV7, Hippo, and PI3K/mTOR, the specific ways in which VD3-GNP causes these effects are not completely understood. Additional examination of the molecular connections and signaling cascades would enhance the biological validity of the findings.

 4. The research is performed using laboratory experiments conducted outside of a living organism, which, although useful, does not fully capture the intricacies of tumor behavior within the human body. Conducting in vivo investigations in animal models could offer further understanding of the treatment's effectiveness, distribution throughout the body, and safety characteristics.

 5. The study does not investigate the potential long-term consequences of VD3-GNP treatment, such as the development of resistance mechanisms in cancer cells. Longitudinal research can provide insight into the long-term sustainability of treatment effects and the development of resistance patterns.

 6. While the study acknowledges the use of non-cytotoxic doses, it is important to conduct a thorough analysis of toxicity, including potential off-target effects and the influence on normal cells, in order to assess the safety of VD3-GNP.

7. Conducting a comparison between VD3-GNP and current breast cancer treatments/chemo could offer a more distinct comprehension of its possible benefits or synergistic effects. This comparative data could provide valuable guidance for incorporating into existing therapy paradigms.

8. The study lacks information on the pharmacokinetic features of VD3-GNP, including its absorption, distribution, metabolism, and excretion. These details are essential for comprehending how the substance behaves in the body.

In addition, I observed multiple grammatical mistakes however, they can be easily rectified; few of them like;

1. "Metastasis in breast cancer is the major cause of female death (about 30%)." - It would be more accurate to say, "Metastasis in breast cancer is the major cause of death in females (about 30%)."

 2. "Our VD3-GNP nanoparticle treatment of breast cancer cells (MCF-7 and MDA-MB-231) significantly reduces the aggressiveness (cancer cel migration and invasion rates >45%)..." - It should be "cancer cell migration..."

3. Consistency in terms like "VD3-GNPs" and "VD3-GNP." Ensure that the terminology remains consistent throughout the paper unless they refer to different things.

4. In scientific writing, it's better to avoid using parentheses for essential information, like "(three orders of magnitude lower than earlier studies)." Instead, it could be integrated into the sentence: "...is surprising, as it is three orders of magnitude lower than earlier studies have shown."

5. "Based on our earlier observation that Vitamin D3 downregulates mTOR we hypothesized that Vitamin D3 con- jugated to gold nanoparticles (VD3-GNP) reduces breast cancer aggressiveness..." - This sentence could use a comma after "mTOR" for clarity: "Based on our earlier observation that Vitamin D3 downregulates mTOR, we hypothesized..."

6. Check the alignment and hyphenation in the text, as some lines appear to be broken up incorrectly, e.g., "con- jugated to gold nanoparticles (VD3-GNP) reduces breast cancer aggressiveness..."

Comments on the Quality of English Language

1. "Metastasis in breast cancer is the major cause of female death (about 30%)." - It would be more accurate to say, "Metastasis in breast cancer is the major cause of death in females (about 30%)."

 2. "Our VD3-GNP nanoparticle treatment of breast cancer cells (MCF-7 and MDA-MB-231) significantly reduces the aggressiveness (cancer cel migration and invasion rates >45%)..." - It should be "cancer cell migration..."

3. Consistency in terms like "VD3-GNPs" and "VD3-GNP." Ensure that the terminology remains consistent throughout the paper unless they refer to different things.

4. In scientific writing, it's better to avoid using parentheses for essential information, like "(three orders of magnitude lower than earlier studies)." Instead, it could be integrated into the sentence: "...is surprising, as it is three orders of magnitude lower than earlier studies have shown."

5. "Based on our earlier observation that Vitamin D3 downregulates mTOR we hypothesized that Vitamin D3 con- jugated to gold nanoparticles (VD3-GNP) reduces breast cancer aggressiveness..." - This sentence could use a comma after "mTOR" for clarity: "Based on our earlier observation that Vitamin D3 downregulates mTOR, we hypothesized..."

6. Check the alignment and hyphenation in the text, as some lines appear to be broken up incorrectly, e.g., "con- jugated to gold nanoparticles (VD3-GNP) reduces breast cancer aggressiveness..."

Author Response

  1. This study specifically examines two types of breast cancer cell lines, namely MDA-MB-231 (triple-negative) and MCF-7 (ER+). While these models have a strong foundation, incorporating a broader spectrum of breast cancer subtypes would yield a more comprehensive comprehension of the treatment's efficacy across various disease profiles. 

Please note that our significant original contributions are:

  1. VD3-GNP treatment is proven to reduce breast cancer aggressiveness; hence this is a possible preferred therapy.
  2. VD3-GNP treatment downregulates key signaling pathways (PI3K/AKT/mTOR, HIPPO, and ETV7) – potentially critical for reducing cancer aggressiveness.
  3. VD3 conjugated with GNP – at a non-toxic nanomolar range of VD3 – delivers three orders of magnitude less VD3 than all previous studies. This likely occurs due to GNP-mediated efficient cellular delivery.

Having established these contributions in two major cell lines should be viewed as adequate proof of concept; papers using only two cell lines are not rare (e.g. new References 70-74). More cell lines would of course be desirable; however, we do not have access to more nor funding (emphasized in the Limitations section). 

The new references are:

  1. Theodossiou, T.A., Ali, M.,Grigalavicius, M. et al.Simultaneous defeat of MCF7 and MDA-MB-231 resistances by a hypericin PDT–tamoxifen hybrid therapy. npj Breast Cancer, 2019, 5, 13 (). https://doi.org/10.1038/s41523-019-0108-8.
  2. Lunetti, P., Giacomo, M. D, Vergara, D., Domenico, S. De., Maffia, M. Zara, V., Loredana Capobianco, L., A Ferramosca. A., Metabolic reprogramming in breast cancer results in distinct mitochondrial bioenergetics between luminal and basal subtypes. The FEBS Journal , 2019,286, 688–709.
  3. Gest, C., Joimel, U., Huang, L. et al.Rac3 induces a molecular pathway triggering breast cancer cell aggressiveness: differences in MDA-MB-231 and MCF-7 breast cancer cell lines. BMC Cancer. 2013,  13, 63 (). https://doi.org/10.1186/1471-2407-13-63.
  4. Tripathi, V., Jaiswal, P., Verma, R., Sahu,K., Shovan Kumar Majumder,S.K., Chakraborty, S., Jha, H. C.,   Singh Parmar, H. S.Therapeutic influence of simvastatin on MCF-7 and MDA-MB-231 breast cancer cells via mitochondrial depletion and improvement in chemosensitivity of cytotoxic drugs, Advances in Cancer Biology – Metastasis, 2023, 9, 2667-3940.
  5. Jia, T., Zhang, L., Duan, Y. et al.The differential susceptibilities of MCF-7 and MDA-MB-231 cells to the cytotoxic effects of curcumin are associated with the PI3K/Akt-SKP2-Cip/Kips pathway. Cancer Cell Int.2014, 14, 126.

  1. Although the study demonstrates the efficacy of VD3-GNP at nanomolar concentrations, the precise range of dose and duration of therapy that is most suitable for clinical use is yet uncertain. Further dose-response studies could assist in establishing more conclusive therapeutic ranges.

We fully agree that for future clinical study, the dosage of VD3-GNP needs to be standardized (based on patients’ demographic criteria) and currently, we don’t have the resources to do it – already addressed in the Limitations section

3. While the study shows a decrease in the activity of important pathways and proteins like ETV7, Hippo, and PI3K/mTOR, the specific ways in which VD3-GNP causes these effects are not completely understood. Additional examination of the molecular connections and signaling cascades would enhance the biological validity of the findings.

Thank you. Our thinking has been quite similar, and we are pursuing this with a scope that goes far beyond the current paper. We feel that the current paper stands on its own and that there is no justification for delaying publication any further.

4. The research is performed using laboratory experiments conducted outside of a living organism, which, although useful, does not fully capture the intricacies of tumor behavior within the human body. Conducting in vivo investigations in animal models could offer further understanding of the treatment's effectiveness, distribution throughout the body, and safety characteristics.

We do not disagree with the referee; of course, this observation applies to every in vitro study. Due to our limited funding resources (mentioned clearly in the Limitations section), we are unable to add in vivo animal investigations as well as human studies. Also, to avoid any confusion, we highlighted the point that we are using cells in the title.

  1. The study does not investigate the potential long-term consequences of VD3-GNP treatment, such as the development of resistance mechanisms in cancer cells. Longitudinal research can provide insight into the long-term sustainability of treatment effects and the development of resistance patterns.

    The referee is obviously correct – but such long-term studies – already being carried out by us, are beyond the scope of the current paper.
  2. While the study acknowledges the use of non-cytotoxic doses, it is important to conduct a thorough analysis of toxicity, including potential off-target effects and the influence on normal cells, in order to assess the safety of VD3-GNP.

       Again, we do not disagree with the referee, but such long-term studies seem unwarranted. It has already been shown that very high doses (up to 200 uM) of VD3 causes cytotoxicity in normal human cells (see References 63, 65). Also recently shown is that GNP can be cytotoxic to TNBC cell lines at high concentrations (25 μg/mL to 1 mg/mL for 24 h), which is higher than the concentration we used (References 66-67). Due to these prior findings, we did not pursue cytotoxicity studies. Also, as mentioned above, our limited funding does not permit us to use normal cell lines currently.

The new references added are:

65.Veeresh, P. K. M., Basavaraju, C. G., Dallavalasa, S., Anantharaju, P. G., Natraj, S. M., Sukocheva, O. A., Madhunapantula, S. V. Vitamin D3 Inhibits the Viability of Breast Cancer Cells In Vitro and Ehrlich Ascites Carcinomas in Mice by Promoting Apoptosis and Cell Cycle Arrest and by Impeding Tumor Angiogenesis. Cancers (Basel). 2023, 15:4833.

66. Lee, J., Chatterjee, D. K., Lee, M. H., Krishnan, S. Gold nanoparticles in breast cancer treatment: promise and potential pitfalls. Cancer Lett. 2014.347:46-53.

67. Surapaneni, S.K., Bashir, S., Tikoo, K. Gold nanoparticles-induced cytotoxicity in triple negative breast cancer involves different epigenetic alterations depending upon the surface charge. Sci Rep.2018, 8: 12295.

  1. Conducting a comparison between VD3-GNP and current breast cancer treatments/chemo could offer a more distinct comprehension of its possible benefits or synergistic effects. This comparative data could provide valuable guidance for incorporating into existing therapy paradigms.

    We are planning these studies with two chemotherapies (Doxorubicin and 5-Fluorouracil); however, this is not within the scope of the current paper.

  2. The study lacks information on the pharmacokinetic features of VD3-GNP, including its absorption, distribution, metabolism, and excretion. These details are essential for comprehending how the substance behaves in the body.

    We agree with the referee that for future clinical studies, pharmacokinetics approaches will be necessary; however, our laboratory is limited to in vitro studies only, and animal/human studies are not possible, nor planned.

In addition, I observed multiple grammatical mistakes however, they can be easily rectified; few of them like;

  1. "Metastasis in breast cancer is the major cause of female death (about 30%)." - It would be more accurate to say, "Metastasis in breast cancer is the major cause of death in females (about 30%)."

    Thanks for the suggestion but our sentence “Metastasis in breast cancer is the major cause of female death (about 30%)." is more succinct, is grammatically correct, and has the same meaning so we did not alter that sentence.

  2. "Our VD3-GNP nanoparticle treatment of breast cancer cells (MCF-7 and MDA-MB-231) significantly reduces the aggressiveness (cancer celmigration and invasion rates >45%)..." - It should be "cancer cell migration..."

    This point has been corrected. Now it is

          “cancer cell migration and invasion rates >45%".

  1. Consistency in terms like "VD3-GNPs" and "VD3-GNP."Ensure that the terminology remains consistent throughout the paper unless they refer to different things.

VD3-GNP is singular, and VD3-GNPs are plural, and have been used correctly, so no change is required.

4. In scientific writing, it's better to avoid using parentheses for essential information, like "(three orders of magnitude lower than earlier studies)." Instead, it could be integrated into the sentence: "...is surprising, as it is three orders of magnitude lower than earlier studies have shown."

"Based on our earlier observation that Vitamin D3 downregulates mTOR we hypothesized that Vitamin D3 conjugated to gold nanoparticles (VD3-GNP) reduces breast cancer aggressiveness..." - This sentence could use a comma after "mTOR" for clarity: "Based on our earlier observation that Vitamin D3 downregulates mTOR, we hypothesized..."

Thanks for the suggestion though we feel there is no error (grammatical or scientific) in this part. We have also implemented your comma suggestion.

5.Check the alignment and hyphenation in the text, as some lines appear to be broken up incorrectly, e.g., "con- jugated to gold nanoparticles (VD3-GNP) reduces breast cancer aggressiveness..."

We have corrected this.

Reviewer 4 Report

Comments and Suggestions for Authors

The authors propose an alternative treatment for triple-negative cancer cells by utilizing gold nanoparticles conjugated with vitamin D3 as a nanosystem to reduce the invasiveness of these cells through alteration of key proteins in the PI3K/AKT/mTOR, Hippo, and ETV7 pathways. While the work is interesting, there are several aspects of the manuscript that require significant improvement before potential publication.

Major revision

1. The introduction requires substantial improvement as it lacks cohesion and fails to follow a common thread, making it challenging to understand the intended objective of this work.

2. Regarding the TEM images, the authors should include a size distribution histogram to provide a comprehensive analysis of the nanoparticles´ size distribution.

3. Figure 2 (FT-IR) present a poor quality, and the absorbance peaks should be included in both spectra to facilitate a more effective comparison. Additionally, brown is not clearly distinguishable from black in the figure.

4. Table 1 is quite confusing due to the use of undefined acronyms. The authors should provide the DLS diameters, along with the standard deviation, for both the gold nanoparticles and the vitamin D3-conjugated gold nanoparticles. Additionally, reporting the polydispersity index is necessary to determine whether the proposed system is monodisperse.

5. Usually, the ZetaSizer (Malvern) equipment is utilized to measure the hydrodynamic diameter. Since the authors have employed different measurement equipment in this case, it is important for them to explain what the primary and secondary axes in Figure 3A correspond to (% passing and % channel, respectively). Furthermore, it is essential to compare the sizes of gold nanoparticles and those conjugated with vitamin D3. These results should be compared and analyzed in relation to those obtained from TEM.

6. The criterion for selecting the concentrations of 1, 2, 5, and 10 mM for vitamin D3 is not provided, leaving it unclear why these specific concentrations were chosen. Additionally, the reason for observing a concentration of 388nM in 100 μL of the sample is not explained, which may raise confusion. Furthermore, the rationale behind choosing a concentration of 38nM for conducting cell viability studies is not clarified. It is important to know whether previous lethal dose 50 studies were conducted to justify this concentration.

7. The process by which the authors have conjugated vitamin D3 to gold nanoparticles is not clearly described in the manuscript. Specifically, it is unclear how the authors determined the conjugation efficiency, i.e., the amount of vitamin D3 that has actually been adsorbed onto the nanoparticles out of the initial concentration of 10 μg/mL. This lack of clarity is particularly noticeable given that the zeta potential for the VD3-GNP system is more negative (-24.18 mV) compared to GNPs alone (-17.5 mV). Additionally, the rationale for choosing the proportion of gold nanoparticles relative to that of vitamin D3 is not justified. Therefore, further clarification is needed regarding the conjugation process and the rationale behind the chosen proportions of gold nanoparticles and vitamin D3.

8. With regard to Figure 4, it is recommendable to conduct tests at 48 and 72 hours to ensure that the proposed system is genuinely non-cytotoxic. Additionally, it would be advisable to include a healthy cell line as a control for comparison.

9. In the discussion of results, the authors claim that the nanoparticles obtained range between 10-15 nm. However, according to the size distribution in Figure 3A, the majority size appears to be approximately 50 nm.

10. The conclusions need improvement, with a focus on highlighting the most significant findings of this research.

Minor revision

1. Due to the large number of acronyms used, the authors should include a list of abbreviations for clarity and ease of understanding.

2. Figures 7, 8, 9 and 10 have very poor quality (they are not readable), therefore the authors should improve them.

3. Line 171 should be revised to replace "polymer nanoparticles" with "gold nanoparticles," as the authors do not work with polymeric nanoparticles.

4. Line 617 should be revised to delete "Supplementary Materials" since the authors do not provide supplementary material.

5. The bibliography lacks a homogeneous format.

Author Response

Major revision

  1. The introduction requires substantial improvement as it lacks cohesion and fails to follow a common thread, making it challenging to understand the intended objective of this work.

    We have shortened the Introduction and improved articulation to address this issue and added subheadings to make it easier to follow.
  2. Regarding the TEM images, the authors should include a size distribution histogram to provide a comprehensive analysis of the nanoparticles´ size distribution.

         New TEM images and a size distribution histograms have now been added   in Figures 2C-J. Thank you for the suggestions.

  1. Figure 2 (FT-IR) presents a poor quality, and the absorbance peaks should be included in both spectra to facilitate a more effective comparison. Additionally, brown is not clearly distinguishable from black in the figure.

We edited the FT-IR spectrum color, added a GNP FT-IR spectrum, and labeled the peaks (Figure 3B). Also, the brown color is now replaced by green to accommodate the reviewer.

  1. Table 1 is quite confusing due to the use of undefined acronyms. The authors should provide the DLS diameters, along with the standard deviation, for both the gold nanoparticles and the vitamin D3-conjugated gold nanoparticles. Additionally, reporting the polydispersity index is necessary to determine whether the proposed system is monodisperse.

We have modified Table 1 as per the reviewer’s suggestion The DLS diameter of GNP is 27.50 nm, and VD3-GNP is 34.60 nm (see Table 1). The GNP PDI index is 1.505, and VD3-GNP PDI index is 1.522 (Table 1).

5.Usually, the ZetaSizer (Malvern) equipment is utilized to measure the hydrodynamic diameter. Since the authors have employed different measurement equipment in this case, it is important for them to explain what the primary and secondary axes in Figure 3A correspond to (% passing and % channel, respectively). Furthermore, it is essential to compare the sizes of gold nanoparticles and those conjugated with vitamin D3. These results should be compared and analyzed in relation to those obtained from TEM.

We have now addressed this issue  (in section 4.2.4) and we have moved Figure 3A  to Supplementary Figure 1 – where we marked the % passing and % channel to address the reviewer’s concern (although the Microtrac  Zetatrac instrument has no primary or secondary axes). We added new TEM size distributions in Figures 2I, J. Also the comparison of the two results have been addressed in the Discussion: "In contrast, our VD3-GNPs’ diameters are in the range of ~14-19 (avg ~ 16nm)  nm (obtained from TEM, diameter ~ 35 nm from DLS, while the GNP diameter ~ 26 nm from DLS; see Table 1); our VD3-GNPs are highly effective – even with a low concentration of VD3 –potentially enabling more efficient cellular uptake and anti-cancer action [47-52]."

6. The criterion for selecting the concentrations of 1, 2, 5, and 10 mM for vitamin D3 is not provided, leaving it unclear why these specific concentrations were chosen. Additionally, the reason for observing a concentration of 388nM in 100 μL of the sample is not explained, which may raise confusion. Furthermore, the rationale behind choosing a concentration of 38nM for conducting cell viability studies is not clarified. It is important to know whether previous lethal dose 50 studies were conducted to justify this concentration.

We didn’t use (or mention) "1, 2, 5, 10 mM” concentrations of VD3 to make our VD3-GNPs in our manuscript so we are not sure where the reviewer found these numbers and we do not understand the basis of this question. In fact, we used 38 nM for the majority of our cellular experiments which was selected on the basis of our MTT experiments (see Methods Section 4.3.2 and Section 2.2.1, Figure 5), that was already provided in the previously submitted manuscript. Additionally, we performed our MTT experiments using several concentrations (starting from 10 ul  of the stock and its serial dilution (1:10, 1:100, 1:1000) for assessment of VD3-GNP cytotoxicity. We have updated the non-toxic dosage details in the revised manuscript (see section 4.3.2):

“Cytotoxicity evaluation of VD3-GNP was performed using the MTT assay [60-61]. Approximately 1 × 105 cells ml−1 (MDA-MB-231 and MCF-7) in their exponential growth phase were seeded in flat-bottomed 96-well polystyrene plates and incubated for 24 hrs. at 37 0C in a 5% CO2 incubator. In the medium, a series of VD3-GNP dilutions (10 μg/ml with 100 μl per well, 1:10, 1:100, 1:1000 from 10 μg/ml stock) were added to the plate in triplicate. After 24 hours of incubation, 50 μl of MTT reagent was added to each well and further incubated at 37 0C in the dark for 2 hrs. The formazan crystals that formed after 2 hrs. of incubation in each well were dissolved in 150 μl of DMSO, and each well was read with a spectrophotometer/fluorometer (DeNovix, DS11 FX, USA) using the nanodrop function at 570 nm with background correction at 630 nm. Wells with complete medium and nanoparticles, MTT reagent, and without cells were used as blanks.”

Also, we measured the VD3 concentration with HPLC in our VD3-GNP which is 38 nM (Figures 4A-B and Section 2.2.6.: HPLC Quantitation of VD3 in VD3-GNP, Page 10, lines 266-274, and page 10 lines 275-276). For details of the HPLC method see Section 4.2.6.: Quantitation of the Vitamin D3 in VD3-GNPs by HPLC (page 27 line 629-page 28).

  1. The process by which the authors have conjugated vitamin D3 to gold nanoparticles is not clearly described in the manuscript. Specifically, it is unclear how the authors determined the conjugation efficiency, i.e., the amount of vitamin D3 that has actually been adsorbed onto the nanoparticles out of the initial concentration of 10 μg/mL. This lack of clarity is particularly noticeable given that the zeta potential for the VD3-GNP system is more negative (-24.18 mV) compared to GNPs alone (-17.5 mV). Additionally, the rationale for choosing the proportion of gold nanoparticles relative to that of vitamin D3 is not justified. Therefore, further clarification is needed regarding the conjugation process and the rationale behind the chosen proportions of gold nanoparticles and vitamin D3.

We used the 10 ul dosage for most of this study so we performed HPLC on that dosage only, which suggests that the Vitamin D3 concentration in the nano formulation (10 ul) is 38 nM (page 10 lines 266-274, page 11 line 275, and Figures 4 A, B), and also added Figure 11 A-D panels to address the synthesis details. Furthermore, we added two new references (References 68-69) where surface modification by different biological agents with different concentrations alters the nanoparticle size and zeta potential. In our study, we used a similar synthesis procedure with a minor modification as stated in Reference 51 (which was already mentioned) and the details of our synthesis method are provided in Section 4.2. Synthesis of GNPs and VD3-GNPs (page 23 lines 590-600, page 24 lines 601-624).

New references:

  1. Wang, W., Ding, X., Xu, Q., Wang, J., Wang, L., Lou, X. Zeta-potential data reliability of gold nanoparticle biomolecular conjugates and its application in sensitive quantification of surface absorbed protein. Colloids and Surfaces B: Biointerfaces, 2016, 148; 541-548.
  2. Khongkow, M., Yata, T., Boonrungsiman, S. et al. Surface modification of gold nanoparticles with neuron-targeted exosome for enhanced blood–brain barrier penetration. Sci Rep, 2019, 9, 8278.

  1. With regard to Figure 4, it is recommendable to conduct tests at 48 and 72 hours to ensure that the proposed system is genuinely non-cytotoxic. Additionally, it would be advisable to include a healthy cell line as a control for comparison.

Adding further time points and cell lines is outside of the scope of this study but is being addressed by our current work. Vitamin D3 toxicity was addressed in detail in the Introduction as well as the Discussion and three new References (65-67) have now been added:

  1. Veeresh, P. K. M., Basavaraju, C. G., Dallavalasa, S., Anantharaju, P. G., Natraj, S. M., Sukocheva, O. A., Madhunapantula, S. V. Vitamin D3 Inhibits the Viability of Breast Cancer Cells In Vitro and Ehrlich Ascites Carcinomas in Mice by Promoting Apoptosis and Cell Cycle Arrest and by Impeding Tumor Angiogenesis. Cancers (Basel). 2023, 15:4833.
  2. Lee, J., Chatterjee, D. K., Lee, M. H., Krishnan, S. Gold nanoparticles in breast cancer treatment: promise and potential pitfalls. Cancer Lett. 2014.347:46-53.
  3. Surapaneni, S.K., Bashir, S., Tikoo, K. Gold nanoparticles-induced cytotoxicity in triple negative breast cancer involves different epigenetic alterations depending upon the surface charge. Sci Rep.2018, 8: 12295.
  4. The conclusions need improvement, with a focus on highlighting the most significant findings of this research.

We have edited our conclusion and it is added below;

“We have established that the VD3-GNP treatment (24 hrs.) significantly reduces the breast cancer cells’ (MDA-MB-231 and MCF-7) invasiveness by altering the key cellular signaling pathway proteins of the PI3K/AKT/mTOR and Hippo pathways and ETV7. Hence VD3-GNP is a strong candidate for clinical therapy. Our future studies will target the gene expression profiles and additional regulatory proteins in the PI3K, Hippo, and other pathways (such as autophagy, apoptosis, etc.)"

 Minor revision

  1. Due to the large number of acronyms used, the authors should include a list of abbreviations for clarity and ease of understanding.
    We have added an acronym list.
  2.  
  3. Figures 7, 8, 9 and 10 have very poor quality (they are not readable), therefore the authors should improve them.
    The figures have been edited to address this issue.
  4. Line 171 should be revised to replace "polymer nanoparticles" with "gold nanoparticles," as the authors do not work with polymeric nanoparticles.

    We have addressed this issue and replaced the polymeric nanoparticle with gold nanoparticle and the line now reads: “To identify the molecular distribution of the vitamin D3 and VD3-GNP in the molecular scaffold of the gold nanoparticles, we recorded the infrared absorbance spectra using a Fourier Transformation Spectrometer (Bruker Scientific, LLC, Billerica, MA, USA) (Figure 3B)."
  5. Line 617 should be revised to delete "Supplementary Materials" since the authors do not provide supplementary material.
    A supplement was submitted with our previous manuscript; a slightly modified version is now re-attached.
  6. The bibliography lacks a homogeneous format.

    We have edited the bibliographic formatting and addressed this issue.

Round 2

Reviewer 1 Report

Comments and Suggestions for Authors

In the intro section, there was some text in red. After fixing this issue, I recommend that this manuscript be accepted for publication in the IJMS journal.

Author Response

Thanks for the suggestion. We have edited the Introduction and incorporated objectives at the suggestion of another referee.

Reviewer 2 Report

Comments and Suggestions for Authors

The author responded very well and addressed all the reviewers' comments. The revised manuscript is well-written and has been edited. However, it can be improved by using consistent font sizes, styles, and color schemes across all figures.
Minor comments:
There is inconsistent spacing in some sentences.

Author Response

The author responded very well and addressed all the reviewers' comments. The revised manuscript is well-written and has been edited. However, it can be improved by using consistent font sizes, styles, and color schemes across all figures.

Thanks for the suggestion. We have modified Figure 10’s image panes and made the same style and font size as the rest of the manuscript.
Minor comments:
There is inconsistent spacing in some sentences.

Thanks for the suggestion. We have now made the spacing between words uniform.

Reviewer 3 Report

Comments and Suggestions for Authors

The authors have worked out on the comments raised.

Author Response

Thank You.

Reviewer 4 Report

Comments and Suggestions for Authors

After reviewing the revised version of the manuscript, the authors should address the following issues prior to potential publication.

Major concern

1. The “Summary” section of the introduction currently reflects the conclusions that the manuscript should present. Consequently, I recommend replacing the current conclusions with this section. Furthermore, authors in the introduction should clearly and concisely define the objectives of the manuscript.

2. The polydispersity index (PDI) serves as a measure of the size distribution within a sample, such as nanoparticles in this case. PDI values range from 0.0 for a perfectly uniform sample to 1.0 for a highly polydisperse sample with various particle sizes. However, the authors have reported PDI values greater than 1, specifically for diameters of 34nm (VD3-GPN, PDI: 1.622) and 27nm (GNP, PDI: 1.505). This indicates significant polydispersity in the samples, which is generally not ideal for such systems. Moreover, it is important to note that PDI values typically max out at 1.0. Therefore, the authors should provide an explanation for these values.

Minor concern

1. The quality of Figure 10 must be further improved, because it is still not readable.

2. There are still typographical errors present in the bibliography.

Author Response

  1. The “Summary” section of the introduction currently reflects the conclusions that the manuscript should present. Consequently, I recommend replacing the current conclusions with this section. Furthermore, authors in the introduction should clearly and concisely define the objectives of the manuscript.

A segment (highlighted yellow) from the Introduction has been compressed (by 60%) and used as the conclusion (in blue text). Furthermore, we have added an objective at the end of the Introduction, heeding your advice.

2. The polydispersity index (PDI) serves as a measure of the size distribution within a sample, such as nanoparticles in this case. PDI values range from 0.0 for a perfectly uniform sample to 1.0 for a highly polydisperse sample with various particle sizes. However, the authors have reported PDI values greater than 1, specifically for diameters of 34nm (VD3-GPN, PDI: 1.622) and 27nm (GNP, PDI: 1.505). This indicates significant polydispersity in the samples, which is generally not ideal for such systems. Moreover, it is important to note that PDI values typically max out at 1.0. Therefore, the authors should provide an explanation for these values.

The polydispersity indexes (PDI) of our samples (VD3-GNP and GNP) are > 1 which is likely due to their tendency to aggregate. That is evident from the presence of some large particles in our TEM size distribution (Figures 2I, J). Also, recent research on VD3 in bone tissue engineering (the reference is added below) clearly suggests that PEG particles can aggregate and that probably causes our VD3-PEG to aggregate, increasing the PDI in VD3-GNP preparation.

“Through the use of PEG, particles will aggregate via steric stabilization and increase stability during storage and application of the system." (See the reference below).

To avoid the aggregation and maintain the homogeneity of our nanoparticle solution we sonicated it for 30 min prior to any experiment. We have added that in our updated discussion section:
"In our nano preparation PDI values of GNP and VD3-GNP >1 (see Table 1), and they have a tendency to aggregate as evidenced by some large particles measured by TEM (Figures 2I, J), so we sonicated the preparation for 30 min before use (to avoid the aggregation)."

Also in the material method section:

“The mixture was filtered with a 0.22μm filter before using it for assays and then sonicated for 30 min."

Reference:

Vu AA, Bose S. Effects of vitamin D3 release from 3D printed calcium phosphate scaffolds on osteoblast and osteoclast cell proliferation for bone tissue engineering. RSC Adv. 2019;9(60):34847-34853. doi: 10.1039/c9ra06630f. Epub 2019 Oct 29. PMID: 35474960; PMCID: PMC9038120.

Minor concern

  1. The quality of Figure 10 must be further improved because it is still not readable.

We have modified Figure 10 by making the individual image panes bigger and we added Supplementary Figure 4 in the previous revision and the current version where each individual PPI network image pane is a full page in size and easy to read.  We hope this will address the reviewer’s issue.

2. There are still typographical errors present in the bibliography.

We have checked our bibliography and removed the errors.

Round 3

Reviewer 4 Report

Comments and Suggestions for Authors

After reviewing the second version of the manuscript, I suggest the following minor revisions:

1. The authors must include the formula used to calculate the polydispersity index in either the materials and methods section or the supplementary material. This index typically ranges between 0 and 1. Therefore, PDI values above 1 not only suggest aggregations but also indicate a completely heterogeneous system.

2. The list of acronyms should be included in the manuscript to facilitate reading comprehension.

Author Response

Minor review:

1.The authors must include the formula used to calculate the polydispersity index in either the materials and methods section or the supplementary material. This index typically ranges between 0 and 1. Therefore, PDI values above 1 not only suggest aggregations but also indicate a completely heterogeneous system.

We have addressed this on page 25 line 640- page 26 line 655, copied below.

“GNP and VD3-GNP particles’ diameters and polydispersity indices were measured by  DLS (Microtrac Zetatrac particle size; Colloid Metrix GmbH, Meerbush, Germany). The GNP and VD3-GNP solutions were diluted (1:10) from the stock for the measurement.

 The polydispersity index is defined as:

PDI= Mw/Mn,

Mw= weight average molecular weight

Mn= Number average molecular weight.

This number is calculated by the Microtrac Zetatrac and is widely used [see, e.g 75, 76]. A PDI = 1 indicates that particles are monodisperse, whereas a PDI larger than 1 suggests a polydisperse sample. We expect that synthetic polymers, like PEG, have PDI > 1. In our experiments, we find PDI ranges up to 1.6, which can result from multi-particle aggregation (further addressed in the Discussion).”

 Further details about the wave mechanics of light and how the instrument measures the quantities are not appropriate for this paper or the readership.

2. The list of acronyms should be included in the manuscript to facilitate reading comprehension.

We have added the acronym list at the end of the manuscript.

We hope that the paper is now totally acceptable, and no further iterations are clearly warranted.